JCB Journal of Cell Biology

# A RAB7A phosphoswitch coordinates Rubicon Homology protein regulation of Parkin-dependent mitophagy

Dan A. Tudorica[1,2,3] , Bishal Basak[1,4] , Alexia S. Puerta Cordova[3,5] , Grace Khuu[1,6] , Kevin Rose[3,5] , Michael Lazarou[1,6] , Erika L.F. Holzbaur[1,4] , and James H. Hurley[1,2,3,7]

Activation of PINK1 and Parkin in response to mitochondrial damage initiates a response that includes phosphorylation of RAB7A at Ser72. Rubicon is a RAB7A binding negative regulator of autophagy. The structure of the Rubicon:RAB7A complex suggests that phosphorylation of RAB7A at Ser72 would block Rubicon binding. Indeed, in vitro phosphorylation of RAB7A by TBK1 abrogates Rubicon:RAB7A binding. Pacer, a positive regulator of autophagy, has an RH domain with a basic triad predicted to bind an introduced phosphate. Consistent with this, Pacer-RH binds to phosho-RAB7A but not to unphosphorylated RAB7A. In cells, mitochondrial depolarization reduces Rubicon:RAB7A colocalization whilst recruiting Pacer to phospho-RAB7A–positive puncta. Pacer knockout reduces Parkin mitophagy with little effect on bulk autophagy or Parkin-independent mitophagy. Rescue of Parkin-dependent mitophagy requires the intact pRAB7A phosphate-binding basic triad of Pacer. Together these structural and functional data support a model in which the TBK1-dependent phosphorylation of RAB7A serves as a switch, promoting mitophagy by relieving Rubicon inhibition and favoring Pacer activation.

## Introduction

Macroautophagy (hereafter autophagy) is an ancient and conserved system whereby cells isolate, degrade, and recycle cytosol, aggregates, and organelles in response to starvation or stress (Yamamoto et al., 2023). Autophagy may take up bulk cytosol non-selectively or may be selective for particular types of cargo (Stolz et al., 2014; Adriaenssens et al., 2022). Autophagy is central to neuronal homeostasis, and dysfunction in autophagy is implicated in neurodegenerative diseases, including Parkinson's disease (PD), Alzheimer's disease, amyotrophic lateral sclerosis (ALS), and frontotemporal degeneration (FTD) (Klionsky et al., 2021). Autophagy protects neurons by clearing aggregation-prone proteins (aggrephagy) and damaged mitochondria (mitophagy) (Uoselis et al., 2023). The PD gene products PINK1 and Parkin represent the best-characterized example of a disease-linked selective autophagy pathway (Themistokleous et al., 2023). The protein kinase PINK1 and the ubiquitin E3 ligase Parkin initiate mitophagy in response to mitochondrial damage (Narendra et al., 2008, 2010). The importance of this process in PINK1/Parkin-associated PD has led to an intense focus on understanding the regulation of PINK1 and Parkin

(Themistokleous et al., 2023), as well as the events of mitophagy initiation that occur immediately downstream of PINK1 and Parkin (Uoselis et al., 2023).

PINK1/Parkin mitophagy initiation entails the activation of Parkin, which catalyzes the addition of short ubiquitin chains to mitochondrial outer membrane proteins. These ubiquitin chains are recognized by selective autophagy adaptors, including NDP52 and OPTN (Heo et al., 2015; Moore and Holzbaur, 2016; Richter et al., 2016). NPD52, OPTN, and other cargo receptors (Vargas et al., 2019; Ravenhill et al., 2019; Turco et al., 2019) then act to recruit the core autophagy machinery, which is broadly conserved among bulk and selective autophagy pathways (Hurley and Young, 2017). The core autophagy machinery includes the Unc-51–like autophagy activating kinase (ULK1) complex, the class III phosphatidylinositol 3-kinase complexes I and II (PI3KC3-C1 and -C2), the PI3P-binding WIPI proteins, the ubiquitin-like ATG8 proteins, the ATG12-ATG5-ATG16L1 complex and associated machinery of ATG8 conjugation, and the ATG2 and ATG9 proteins involved in lipid transfer and autophagosome expansion (Chang et al., 2021; Yamamoto et al.,

[1]Aligning Science Across Parkinson's (ASAP) Collaborative Research Network, Chevy Chase, MD, USA;   [2]Graduate Group in Biophysics, University of California, Berkeley, Berkeley, CA, USA;   [3]California Institute for Quantitative Biosciences, University of California, Berkeley, Berkeley, CA, USA;   [4]Department of Physiology, University of Pennsylvania Perelman School of Medicine, Philadelphia, PA, USA;   [5]Department of Molecular and Cell Biology, University of California, Berkeley, Berkeley, CA, USA;   [6]Walter and Eliza Hall Institute of Medical Research, Melbourne, Australia;   [7]Helen Wills Neuroscience Institute, University of California, Berkeley, Berkeley, CA, USA.

Correspondence to James H. Hurley: jimhurley@berkeley.edu.

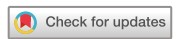

2023). One of the most critical steps in the pathway is the generation of PI3P by PI3KC3. Membrane recruitment of the WIPI proteins is gated by the presence of PI3P (Proikas-Cezanne et al., 2015). The conjugation of the ATG8 family proteins and recruitment of the ATG2 lipid transporter for autophagosome expansion depends on the presence of their cognate WIPI proteins (Dooley et al., 2014; Bakula et al., 2017). This in turn depends entirely on the production of PI3P by PI3KC3.

The centrality of the PI3KC3 complexes to autophagy initiation renders their tight regulation indispensable. PI3KC3-C1 and C2 share the subunits VPS34, VPS15, and BECN1. PI3KC3-C1 uniquely contains the fourth subunit ATG14, while PI3KC3-C2 contains UVRAG. PI3KC3-C1 is uniquely involved in autophagy, while C2 is involved in both endosome biogenesis and autophagy. Rubicon is a negative regulator of PI3KC3-C2 (Zhong et al., 2009; Matsunaga et al., 2009). Rubicon consists of an N-terminal RUN domain (Sun et al., 2011), a central intrinsically disordered region that includes the PI3KC3-C2 binding domain (Chang et al., 2019), and a C-terminal Rubicon Homology (RH) domain responsible for binding to RAB7A (Tabata et al., 2010; Bhargava et al., 2020). Consistent with its role as a negative regulator of a core autophagy complex, Rubicon knockdown or knockout correlates with resistance to neurodegeneration-associated phenotypes. Genetic ablation of Rubicon improves the resistance of mice to the formation of seeded α-synuclein aggregates, which are characteristic of PD, and blunts the locomotive dysfunction that typifies aging in *Drosophila* (Nakamura et al., 2019). Rubicon knockout also improves non-neuronal phenotypes, including resistance to interstitial kidney fibrosis (Nakamura et al., 2019) and nonalcoholic fatty liver disease (Tanaka et al., 2016) in mice. Conversely, the upregulation of Rubicon is associated with neurodegenerative disease. Its expression is upregulated in the spinal cords of sporadic ALS patients (Beltran et al., 2019).

Rubicon is a member of the protein family defined by the presence of a C-terminal RH domain that includes two other autophagy regulators, PLEKHM1 (Tabata et al., 2010) and Pacer (Cheng et al., 2017, 2019). Despite the presence of an RH domain in both proteins, Pacer appears to act in direct opposition to Rubicon, activating PI3KC3-C2 rather than inhibiting it (Cheng et al., 2017, 2019). Elevated Rubicon expression in the spinal cords of sporadic ALS patients is paralleled by reduced Pacer expression (Beltran et al., 2019). Knockdown of Pacer expression via shRNA impairs bulk autophagic flux and the clearance of pathogenic protein aggregates (Beltran et al., 2019). Here, we use "elevated" and "reduced" in a relative sense, as PI3KC3 subunits, Pacer, and Rubicon are all present at low concentrations and copy numbers in cells, in single-digit to low double-digit nanomolar levels (Cho et al., 2022). Thus, Rubicon and Pacer are structurally related proteins that nevertheless appear to play opposing roles in both the cell biology of autophagy and the pathophysiology of neurodegeneration.

RAB7A is recruited to mitochondria following mitochondrial damage in a process regulated by RABGEF1 and TBC1D15/17, a RAB7A GEF and GAP, respectively (Yamano et al., 2018; Hegedűs et al., 2016). RAB phosphorylation has emerged as a major theme in the regulation of membrane traffic and PD, in particular (Pfeffer, 2023). RAB7A is phosphorylated at Ser 72 by Tank-binding kinase 1 (TBK1) and leucine-rich repeat kinase 1 (LRRK1) (Nirujogi et al., 2021; Fujita et al., 2022; Heo et al., 2018). An extensive body of data supports a central role for TBK1 in PINK1/Parkin-dependent mitophagy (Heo et al., 2015; Richter et al., 2016; Moore and Holzbaur, 2016; Vargas et al., 2019; Nguyen et al., 2023). Finally, and most directly, RAB7A Ser72 phosphorylation is necessary for efficient PINK1/Parkin-dependent mitophagy (Heo et al., 2018).

Here, we show that the binding of RAB7A to the RH family proteins Rubicon and Pacer constitutes a bidirectional switch. Under basal conditions, Rubicon binds unphosphorylated RAB7A and antagonizes autophagy. When RAB7A is phosphorylated following mitochondrial depolarization, Rubicon is displaced, Pacer is recruited, and mitophagy is thereby accelerated. This switch is governed by structural differences in the RH domains of Rubicon and Pacer that have evolved to selectively bind dephospho-RAB7A and pSer72-RAB7A, respectively.

## Results

### Rubicon inhibits Parkin-dependent mitophagy

The negative regulatory role of Rubicon in starvation-induced autophagy and xenophagy has been well-characterized (Zhong et al., 2009; Tabata et al., 2010), but it is unknown if Rubicon regulates PINK1/Parkin-dependent mitophagy. Rubicon KO HeLa cells were generated (Fig. S1, A and B) and stably transfected with either LC3B-HaloTag to examine bulk autophagy or HA-Parkin and Su9-GFP-HaloTag to examine Parkin-dependent mitophagy. Activation of autophagy induces the engulfment of these reporters within autophagosomes, trafficking to lysosomes, and degradation. To initiate the assay, cells were pulsed with membrane-permeable fluorescent HaloLigand, which binds HaloTag and renders it resistant to lysosomal processing (Yim et al., 2022). Bulk autophagy was induced by incubation in Earle's Balanced Salt Solution (EBSS). Mitophagy was induced by treatment with the electron transport chain inhibitors Oligomycin and Antimycin A (OA). Autophagy flux was measured as the fraction of the pulsed reporter digested within a given interval.

Rubicon knockout strongly increased the extent of both starvation-mediated (by 55%) and basal autophagy (by 47%) (Fig. 1, A and B), consistent with past reports (Matsunaga et al., 2009; Nakamura et al., 2019). We found that Rubicon knockout also increased mitophagy flux, though to a lesser degree than for starvation-induced autophagy (23% increase in mitophagy flux over 6 h of depolarization) (Fig. 1, C and D). The significantly smaller impact of Rubicon KO on mitophagy flux indicates that there is a mechanism that renders mitophagy more resistant to Rubicon inhibition than starvation-induced autophagy.

### Phosphorylation of RAB7A Ser72 controls binding to RH domains

The RH domain of Rubicon binds to RAB7A, and RAB7A phosphorylation at Ser72 is required for efficient mitophagy flux. We noticed that in the crystal structure of the Rubicon RH:RAB7A-GTP complex, Ser72, part of the GTP-dependent switch II region

**Figure 1.** **Rubicon depletion promotes bulk autophagy and Parkin mitophagy. (A)** LC3-HaloTag processing assay was used to assess starvation autophagy in WT and Rubicon KO HeLa cells. Cells were incubated in either a complete growth medium or EBSS for 3 h prior to collection and analysis. **(B)** Quantification of starvation autophagy flux in A. Autophagy flux was quantified as the fraction of HaloTag signal in the lower, processed band relative to total signal ($n$ = 2 for rich conditions, $n$ = 4 for starvation conditions). Significance scores derived from a one-tailed $t$ test. Error bars indicate standard deviation. **(C)** Su9-GFP-HaloTag processing assay was used to assess mitophagy flux in Parkin expressing WT and Rubicon KO HeLa cells. Cells were either incubated in rich medium or rich medium supplemented with 10 μM Oligomycin A and 5 μM Antimycin A for 6 h. **(D)** Quantification of mitophagy flux in C. Mitophagy flux was quantified as the fraction of HaloTag signal in the lower, processed band relative to total signal ($n$ = 6 for WT and Rubicon KO). Significance scores derived from a one-tailed $t$ test. Error bars indicate standard deviation. All statistical replicates were drawn from independent experiments, and each lane contains lysates from independent experiments. Source data are available for this figure: SourceData F1.

of RAB7A, is in direct contact with a sterically confined region of the RH domain (Bhargava et al., 2020). We therefore asked whether the phosphorylation state of RAB7A modulated Rubicon binding. We also considered the possibility that RAB7A phosphorylation might regulate Pacer since Pacer also contains an RH domain (Fig. 2 A). However, our structural modeling suggested that the analogous region of interaction with RAB7A is less sterically confined.

To probe these interactions, we expressed and purified recombinant Rubicon and Pacer RH domains, RAB7A, and one of the key kinases known to phosphorylate RAB7A, TBK1 (Heo et al., 2018). Incubation of RAB7A with TBK1 in the presence of ATP resulted in stoichiometrically phosphorylated pSer72 RAB7A. Assay of this material using a PhosTag gel, which slows the migration of phosphorylated proteins and resolves distinct phosphorylated and unphosphorylated bands, confirmed that the kinase reaction proceeded essentially to completion (Fig. 2 B). To assay for RAB7A binding by purified RH domains, we used a confocal microscopy-based bead binding assay. Amylose beads were preloaded with MBP-RH domains and incubated in a solution of the Alexa Fluor 647–labeled GTP-locked RAB7A Q67L

(Fig. 2 C). The Rubicon RH domain robustly bound unphosphorylated RAB7A but did not observably bind phosphorylated RAB7A. The opposite was true for the Pacer RH domain. Pacer-RH robustly bound only to Ser72 pRAB7A, with little to no binding observed to unphosphorylated RAB7A (Fig. 2 D). We followed up on these results, measuring the binding affinity of Rubicon RH for RAB7A via isothermal titration calorimetry (ITC). This binding interaction is characterized by a $K_D$ of 1.7 ± 0.3 μM (Fig. 2 F). This represents a moderate affinity interaction, consistent with a regime where phosphorylation-induced affinity changes could effectively regulate the interaction between Rubicon and Pacer-binding competent states for RAB7A.

To attempt to explain why phosphorylation of RAB7A has opposing effects on binding to the Rubicon and Pacer RH domains, homology modeling was used to generate predictions for the Pacer-pRAB7A binding interface (Fig. 2 E). These models were then compared with the crystal structure of the Rubicon RH:RAB7A complex (Bhargava et al., 2020). The Pacer RH domain structure has a prominent basic patch that is absent in the Rubicon RH domain and appears poised to bind the pSer72 phosphate. Basic residues Lys534, Arg623, and Arg628 within

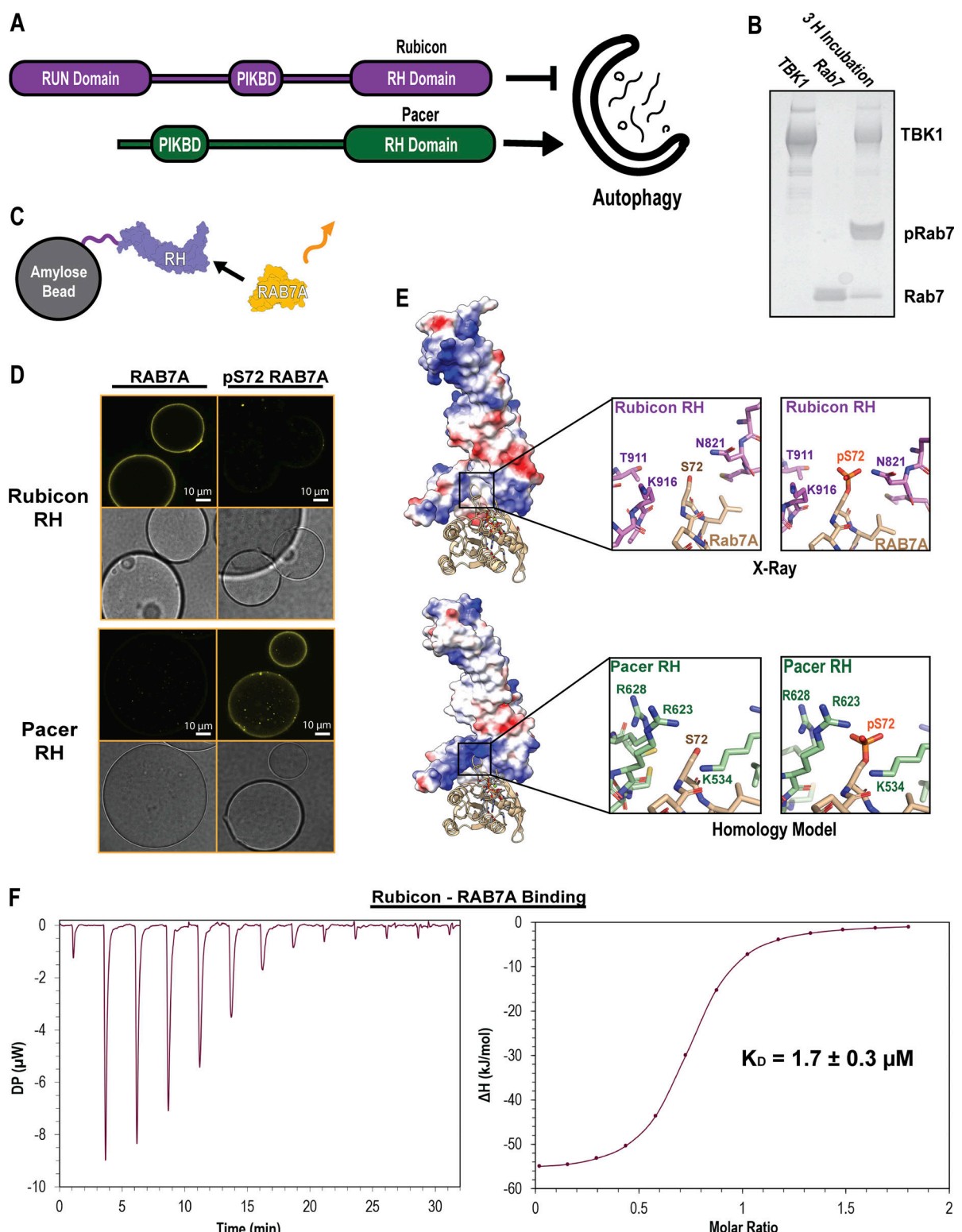

**Figure 2. Rubicon and Pacer RH domain have inverted pRAB7A binding specificity. (A)** Domain maps of Rubicon and Pacer. Rubicon and Pacer both bind UVRAG via their PIK Binding Domains (PIKBD), and both possess RH. **(B)** PhosTag gel showing in vitro phosphorylation of RAB7A using recombinantly purified TBK1. Here, TBK1 was mixed with Q67L, GTP-locked RAB7A for 3 h at room temperature, supplemented with ATP and Mg. PhosTag gels slow the migration of phosphorylated proteins. **(C)** Confocal-based fluorescent bead-binding assay schematic. Amylose beads were functionalized with MBP-RH domain and then incubated in solutions of either phosphorylated or unphosphorylated RAB7A chemically conjugated to a fluorescent dye. **(D)** Amylose beads were incubated in 500 nM MBP-RH domain and 2 µM Alexa Fluor 647647-labeled RAB7A prior to washing and imaging on a confocal setting. Rab7/Rubicon image derived from the same source image as Fig. 3 D. **(E)** Crystal structure of the Rubicon RH domain in complex with RAB7A (top, PDB: 6wcw), and a homology model of the Pacer RH domain in complex with RAB7A (bottom, SwissModel). Coloration represents electrostatic potential (blue = positive charge, red = negative charge).

**(F)** Isothermal Titration Calorimetry (ITC) injection plot (left) and binding curve (right) of Rubicon binding to RAB7A. 13 injections performed at 25°C with 150-s delays between injections. Data were fit to a binding curve assuming a simple, single-binding site mechanism. The binding curve best fit suggests $n = 0.7$. Source data are available for this figure: SourceData F2.

Pacer are all close enough to the negatively charged pSer72 phosphate to form direct salt bridges. In contrast, the Rubicon crystal structure shows that phosphorylation of Ser72 creates a steric clash by introducing the bulk of the phosphate group into a constricted site (Bhargava et al., 2020).

### Phosphorylation of RAB7A Ser72 controls subcellular localization of RH proteins

To determine whether the in vitro binding preferences were correlated with subcellular localization, we transfected Parkin-expressing HeLa cells with mCherry-Rubicon and performed immunofluorescence microscopy using anti-RAB7A and anti-pSer72-RAB7A (Abcam ab302494) (Fig. 3, A and B). In these cells, Rubicon clusters near the nucleus and colocalizes with RAB7A. In contrast, pSer72 RAB7A puncta were found throughout the cytoplasm and colocalized only sporadically with Rubicon. We found that Rubicon is more than threefold more likely to colocalize with RAB7A than with pSer 72 RAB7A, consistent with the in vitro finding that RAB7A phosphorylation negatively regulates Rubicon binding (Fig. 3 C).

### An RH domain basic triad controls pRAB7A binding

To test the structural model for phosphorylation-dependent RAB7A binding, we generated a triple Rubicon RH mutant that contains the Pacer basic triad at the equivalent residues (Rubicon N821K, T911R, K916R), and performed the confocal bead binding assay using these mutants. The Rubicon mutant recruited pRAB7A to beads more effectively than it recruited RAB7A to beads (Fig. 3 D). Overall, it appears that this mutation enabled Rubicon to bind pRAB7A at the cost of a moderate decrease in its ability to bind RAB7A (Fig. 3 E). This suggests that the Pacer basic triad is responsible for conferring binding to pSer72 RAB7A. Reciprocally, mutants of the Pacer RH domain that swap these residues to their Rubicon equivalents (K534N, R623T, R628K) were generated, purified, and assayed in the fluorescent bead binding assay. The recruitment of pRAB7A was sharply diminished by even one of these mutations, while the double and triple mutants resulted in nearly complete loss of pRAB7A binding (Fig. 4, A and B). These mutations did not, however, confer on Pacer the ability to bind dephosphorylated RAB7A.

To determine the role of the RAB7A phosphoswitch on the localization of Pacer in cells, we compared the colocalization of wild-type and mutant Pacer with pRAB7A. Stable Parkin-expressing HeLa cells were transiently transfected with mCherry-tagged Pacer or K534N/R623T Pacer, fixed, and imaged by confocal microscopy (Fig. 4, C and D). Pacer formed sparse punctate structures evenly distributed throughout the cell, in contrast to the perinuclear localization of Rubicon. Pacer colocalized extensively with pRAB7A (40%), while only about 10% of Pacer[K534N/R623T] did so (Fig. 3 E), demonstrating that

these residues are critical for proper Pacer localization. These data show that both Rubicon and Pacer binding to, and colocalization with, RAB7A is controlled by the phosphorylation state of RAB7A Ser72, which is in turn subject to induction by the same conditions that induce mitophagy.

### Pacer regulates PINK/Parkin-dependent mitophagy

Pacer was reported to upregulate starvation-induced autophagy (Cheng et al., 2017), but roles in selective autophagy have not been reported in the literature. To compare the roles of Pacer in bulk autophagy and mitophagy, Pacer knockout HeLa cells were generated via CRISPR-mediated deletion (Fig. S1 C), and the autophagy and mitophagy flux reporters, LC3B- and Su9-GFP-HaloTag, were stably transfected to probe bulk autophagy and mitophagy.

Pacer knockout HeLa cells displayed significantly reduced mitophagy flux following mitochondrial depolarization with OA (Fig. 5, A and B). To probe the contribution of pRAB7A binding to Pacer activity, Parkin and mitophagy reporter-expressing cells were stably transfected with vector, wild-type Pacer, and Pacer[K534N, R623T]. These cells were depolarized for 2 and 4 h, and the mitophagy flux was measured. Rescue with WT Pacer was evident after 2 h, and mitophagy flux after 4 h increased by up to 47% relative to KO cells. In contrast, the Pacer[K534N, R623T] mutant was unable to promote mitophagy flux above the levels seen in Pacer KO cells, suggesting that pRAB7A binding is indispensable to the mitophagy-promoting activity of Pacer (Fig. 5, C and D).

To determine if Pacer might have a role in Parkin-independent mitophagy, these same cells were treated with the iron chelator deferiprone (DFP). DFP-mediated mitochondrial iron loss results in increased expression of mitophagy receptors including NIX and BNIP3 through HIF1a signaling, initiating mitophagy without activating the PINK-PARKIN pathway (Allen et al., 2013; Hara et al., 2020; Zhao et al., 2020). Consistent with this, treatment of cells with DFP does not result in an increase in pS72 Rab7 as measured via immunofluorescence (Fig. S2). There was no discernible difference between Pacer KO cells with vector control or wild-type or mutant Pacer at 8, 16, or 24 h (Fig. 5, E and F). This indicates that Pacer has a specialized, pS72 Rab7-dependent role in PINK1/Parkin-dependent mitophagy.

The impact of Pacer and pRAB7A-dependent action on starvation autophagy was assayed by creating stable transfectants of wild-type Pacer and Pacer[K534N, R623T] in Pacer KO HeLa cells expressing the LC3B-Halo bulk autophagy reporter. In contrast to the previous report of a Pacer role in starvation autophagy (Cheng et al., 2017), these cells showed negligible change in the level of starvation-induced bulk autophagy flux at any time point tested (Fig. 5, G and H). This indicates that, at least in HeLa cells, Pacer is principally an activator of PINK1/Parkin mitophagy rather than of autophagy generally.

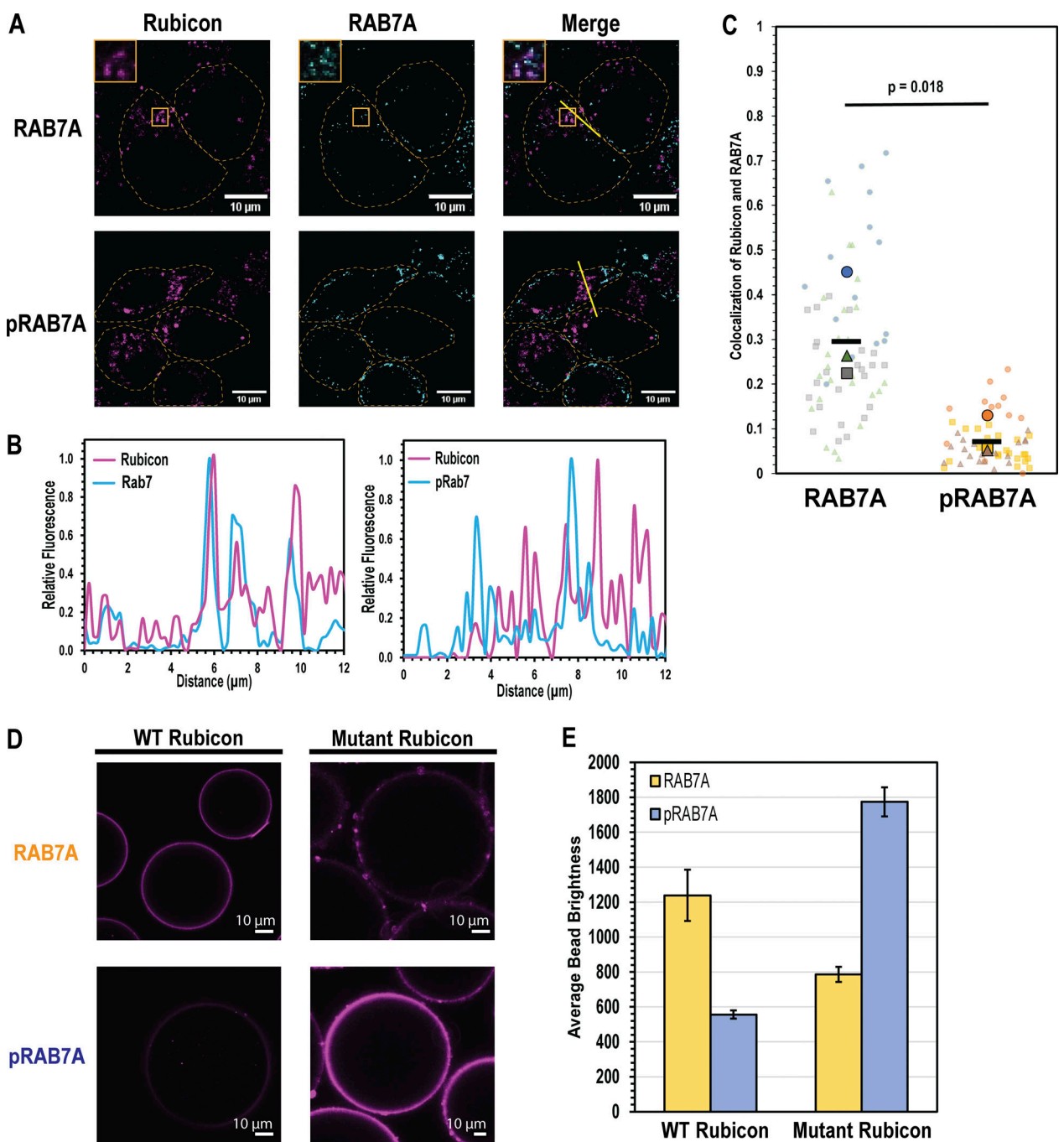

Figure 3. **pRAB7A binding preferences dictate subcellular localization. (A)** WT HeLa cells expressing untagged Parkin were transfected with mCherry-labeled, full-length Rubicon, and then immunofluorescence was performed using general and pS72-specific RAB7A antibodies alongside an Alexa 488 secondary antibody. Rubicon-RAB7A colocalization was examined under rich conditions, and Rubicon-pRAB7A was quantified following 1 h depolarization with 10 μM CCCP. Insets are digitally expanded images, 3.7 μm wide. **(B)** Line scans indicated in RAB7A (left) and pRAB7A (right) imaging trials. Each point on the line scan was baseline subtracted and then normalized to the maximum signal on the line scan for each channel. **(C)** Colocalization was calculated as the above-threshold Mander's colocalization coefficient representing the fraction of Rubicon-positive pixels that are also RAB7A/pRAB7A positive. For each experiment, ∼20 cells were analyzed in this way (transparent markers), and then the average of these 20 measurements constitutes one biological replicate (solid markers). The average of three biological replicates is represented by the horizontal bar marker. The p statistic was calculated using a one-tailed, paired t test comparing average colocalization fractions measured on the same day. **(D)** Confocal bead-binding assay, conducted as in Fig. 2 D. Amylose beads were incubated with either MBP-Rubicon RH or Mutant MBP-Rubicon RH N821K, T911R, and K916R. Rab7/Rubicon image derived from the same source image as Fig. 2 D. **(E)** Quantification of D. Briefly, an ROI was defined on the edge of a bead to be analyzed, and the maximum pixel value in the ROI was recorded as the brightness of the bead. Each condition consists of three independent experiments in which the brightness of ∼20 beads was measured and averaged. Error bars indicate the standard deviation of these independent experiments.

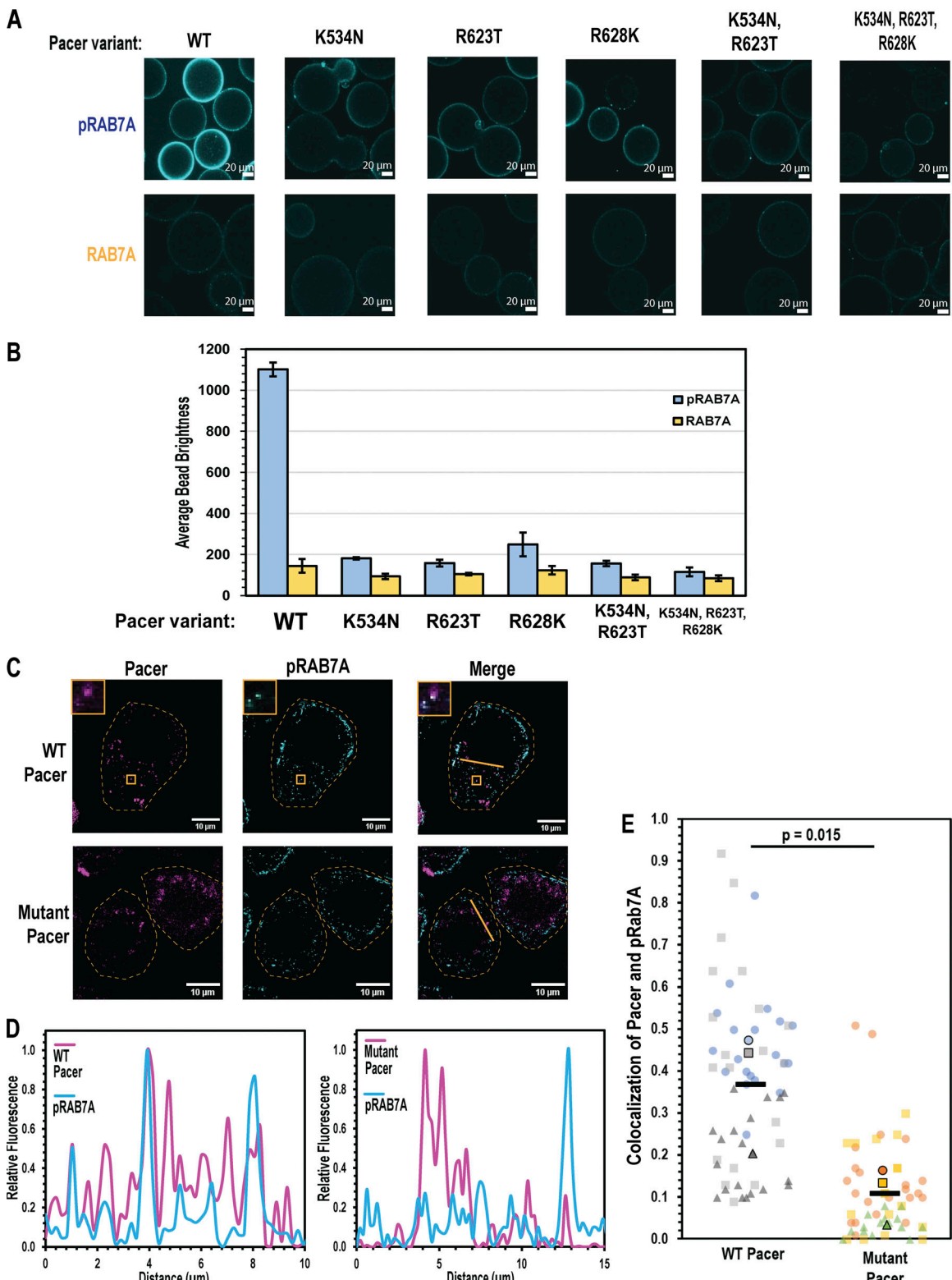

Figure 4. **The Pacer RH domain basic triad controls subcellular localization. (A)** Fluorescent bead binding experiment was performed as in Fig. 2 D and quantified as in Fig. 3 D. Mutants of MBP-Pacer RH with the listed mutant were used in these experiments. **(B)** Quantification of A, conducted as in Fig. 3 E. Each experiment consists of ~20 different beads measured and averaged. This was repeated three times, and the mean of these biological replicates was plotted. Error bars indicate the standard deviation of separate experiments. **(C)** Colocalization of mCherry-wild-type Pacer and mCherry-Mutant (K534N, R623T) Pacer. Cells were transfected with the listed Pacer construct and depolarized for 1 h using CCCP. Immunofluorescence was performed using the pS72 specific antibody of RAB7A, and the Alexa 488 secondary antibody was used to mark pRAB7A. Insets are digitally expanded images, 2.8 µm wide. **(D)** Linescans indicated in C, calculated as in Fig. 3 B. **(E)** Colocalization of WT and Mutant Pacer with pRAB7A, calculated as in Fig. 3 C.

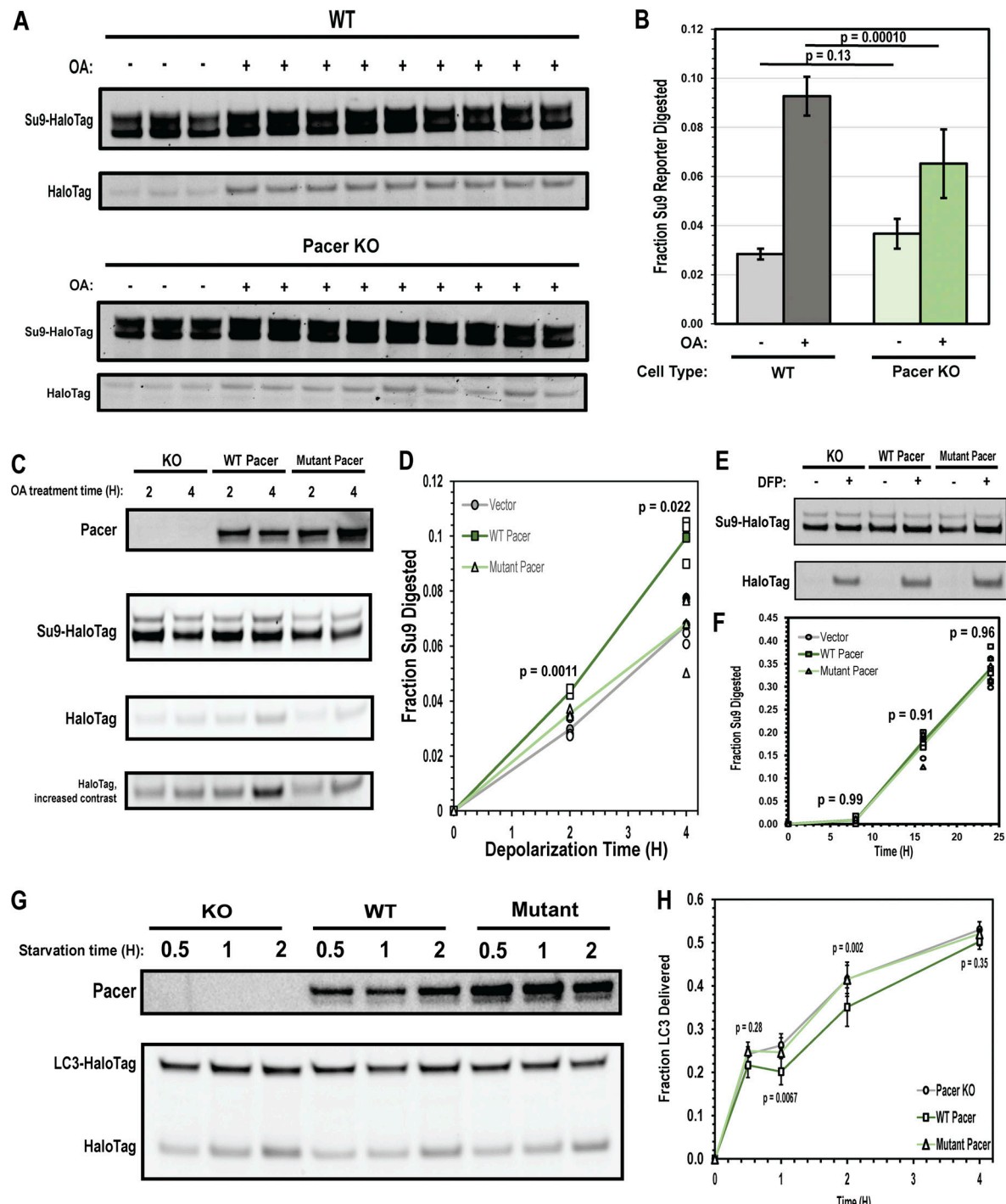

Figure 5. **Pacer is a pS72 RAB7A-dependent activator of mitophagy. (A)** Su9-HaloTag processing assay conducted as in Fig. 1 C. Parkin-expressing cells were treated with either complete medium or complete medium supplemented with 10 µM Oligomycin A, 5 µM Antimycin A for 4 h. Each lane represents lysates from separate experiments. **(B)** Quantification of A, conducted as in Fig. 1 D. n = 3 for the uninduced condition, n = 9 for the induced conditions. Error bars indicate standard deviation and the P value was calculated using a one-tailed t test. **(C)** Pacer KO cells expressing Parkin and Su9-HaloTag reporter were stably transfected with either Pacer, the pRAB7A binding mutant Pacer characterized in Fig. 4, C–E, or empty vector, and then depolarized for 2 or 4 h. n = 3 independent experiments. Pacer expression was probed via Western blot, while HaloTag processing was directly imaged using a fluorescent Halo ligand. **(D)** Quantification of C. P value was calculated using a one-factor ANOVA at 2 and 4 h. **(E)** The same cells used in C were exposed to the non-Parkin mitophagy inducer DFP for 8, 16, and 24 h, and analyzed as with previous Su9-Halo processing blots. **(F)** Quantification of D. n = 4 independent experiments. P value was calculated using a one-factor ANOVA at each time point. **(G)** Pacer KO cells expressing LC3-Halo were stably transfected using either WT Pacer, pRAB7A-binding mutant Pacer, or empty vector, and the starvation autophagy flux assay was conducted as in Fig. 1 A. **(H)** Quantification of F. Between three and nine independent experiments were conducted for each time point. P value was calculated using a one-factor ANOVA at each time point. Source data are available for this figure: SourceData F5.

### Pacer upregulates a prefusion stage of mitophagy

Pacer is thought to potentially promote autophagy via two mechanisms. First, Pacer is known to bind PI3KC3-C2 (Cheng et al., 2017) through a domain also present in Rubicon (Chang et al., 2019). Second, Pacer has been proposed to promote recruitment of the HOPS complex to sites of autophagosome-lysosome fusion (Cheng et al., 2017). HOPS complex recruitment to promote autophagosome–lysosome fusion is also dependent on RAB7A (Takáts et al., 2014; Jiang et al., 2014). While HOPS functions at the last stage in the progression of autophagy, PI3P production is essential throughout the initiation and expansion stages preceding fusion with lysosomes. To determine whether the pRAB7A-binding dependent action of Pacer occurs early or late in mitophagy, we imaged mCherry-tagged wild-type and mutant Pacer-expressing cells using confocal microscopy. We quantified autophagosome formation and autophagosome/lysosome colocalization under nondepolarizing and depolarizing conditions using live-cell imaging. Autophagosomes were monitored with transfected HaloTag-LC3B, and lysosomes were tracked via transfection with LAMP1-mNeon green (Fig. 6 A). Immunofluorescence of the mitochondrial marker HSP60 and LC3B in this system revealed that essentially all autophagosomes formed following depolarization contained mitochondria (Fig. S3).

We found that the number of wild-type Pacer puncta per cell ranged from 2 to 10 puncta under nondepolarizing conditions but increased as much as threefold following mitochondrial depolarization using CCCP. In contrast, the number of mutant Pacer puncta was insensitive to mitochondrial depolarization (Fig. 6 B). Pacer localized robustly with LC3B (Fig. 6 A). Expression of wild-type Pacer during OA treatment increased the total surface area positive for LC3B in the cell by 58% (Fig. 6 C). The total autophagosome–lysosome colocalized area also increased by 70% upon WT Pacer expression (Fig. 6 D), roughly proportionate to the increase in the LC3B positive area alone. This implies that the increased mitophagy flux that characterizes Pacer expression is primarily due to an increase in autophagosome formation rather than delivery to and fusion with lysosomes. If Pacer were principally a promoter of fusion in mitophagy, we would have expected the total LCB3-positive area to decrease as a result of Pacer expression due to accelerated fusion and degradation. Additionally, as an index of autophagosome-lysosome fusion efficiency, we quantified the fraction of the autophagosome area that is also positive for lysosomal markers on a per cell basis. Here, we found that there were only minor differences in fusion efficiency across the KO, wild-type, and mutant conditions. Wild-type Pacer trends towards significance in promoting fusion (P = 0.07), suggesting that Pacer may have fusion-promoting activity, but this likely plays only a minor role in mitophagy promotion in this system (Fig. 6 E).

To further probe the role of Pacer in mitophagy initiation, we cotransfected fluorescently labeled wild-type Pacer and the pRab7-binding deficient Pacer$^{K534N, R623T}$ mutant with Halo-tagged WIPI2 and assessed the degree of WIPI2 puncta formation after treatment with the mitochondrial depolarizers oligomycin/antimycin (Fig. 7 A). WIPI2 is a marker of

autophagosome expansion and marks sites of autophagy initiation. WIPI2 is directly recruited by the lipid product of PI3KC3, PI3P, which is required for phagophore expansion. Transfection with wild-type Pacer more than doubled the WIPI2 punctate signal relative to the vector-only treatment, while transfection with Pacer$^{K534N, R623T}$ failed to increase WIPI2 puncta formation in a statistically significant manner (Fig. 7 B). This confirms that the role of Pacer in autophagy initiation activity is correlated with its ability to promote WIPI2 recruitment to sites of mitophagy initiation. This is consistent with the model that Pacer promotes PI3P production to promote mitophagy initiation. Finally, the number of WIPI2 puncta also positive for WT Pacer exceeded the number of puncta also positive for mutant Pacer more than twofold (Fig. 7 C), confirming that pRab7A binding allows Pacer to localize to sites of mitophagy initiation.

In contrast to wild-type Pacer, transfection with the Pacer$^{K534N, R623T}$ did not significantly increase the LC3B-positive surface area or the LC3B-LAMP1 colocalized area and failed to promote the formation of PI3P as measured via WIPI2 recruitment to nascent autophagosomes. Taken together, this indicates that the formation of pS72 Rab7 at sites of mitophagosome initiation acts to recruit Pacer, which in turn promotes the formation of PI3P and recruitment of WIPI2 and the machinery for ATG8 protein lipidation and autophagosome expansion.

## Discussion

Here, we uncovered an unexpected and elegant mechanism whereby a single RAB phosphorylation event leads to the dissociation of an inhibitor of mitophagy and the recruitment of an activator. The roles of a variety of molecules in mitophagy, including RAB7A, Pacer, and Rubicon, have thus been put into a new perspective. We found that RAB7A is not phosphorylated under starvation conditions. In the absence of RAB7A phosphorylation, Rubicon remains a stronger inhibitor of starvation-mediated autophagy. Pacer, however, appears to be a potent and specific PINK1/Parkin-dependent mitophagy accelerator, whilst having a minimal role in starvation-induced autophagy and DFP-induced mitophagy. This is consistent with the strong dependency of Pacer action on RAB7A phosphorylation. Here, we found that PINK1/Parkin mitophagy increases less in response to Rubicon KO than starvation autophagy. The difference can be explained by the ability of RAB7A phosphorylation to abolish the binding of Rubicon to RAB7A, which removes the Rubicon blockade upon mitophagy induction. Finally, we provide direct evidence that Pacer localizes to sites of mitophagy initiation to promote autophagosome elongation via promoting the production of PI3P.

Phosphorylation of RAB proteins, especially in switch II at residues equivalent to RAB7A Ser72, is a frequent point of regulation in membrane trafficking (Waschbüsch and Khan, 2020; Pfeffer, 2023). The discovery that the LRRK2 kinase mutated in PD acts by phosphorylating RAB proteins has spurred intense efforts to characterize the downstream effects of RAB phosphorylation (Pfeffer, 2023). RAB7A is not a known substrate of LRRK2, rather it appears to be a substrate of at least two other

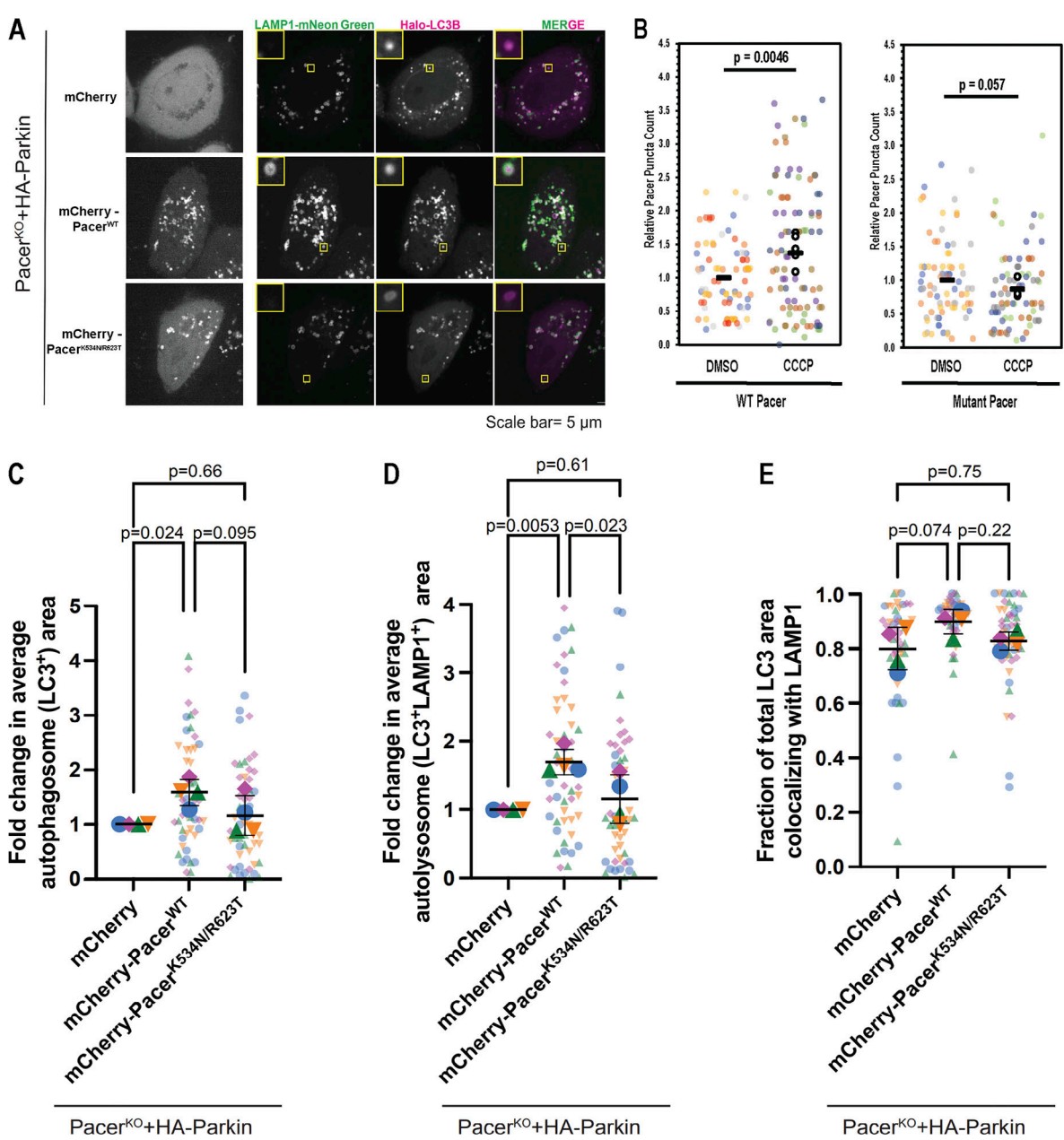

Figure 6. **pRAB7A-dependent function of Pacer in mitophagosome expansion. (A)** Pacer KO HeLa cells stably expressing Parkin were transiently transfected with mCherry-Pacer WT and the pRAB7A binding mutant of Pacer characterized in Fig. 4, C–E. Cells were cotransfected with LAMP1-mNeon as a lysosomal marker and Halo-LC3 as an autophagosome marker. Cells were depolarized for 4 h in 10 μM Oligomycin A and 5 μM Antimycin A prior to live imaging and quantification in C, D, and E. Insets are 4 μm wide. **(B)** Pacer KO HeLa cells stably expressing Parkin and mCherry-WT Pacer/mCherry-Mutant Pacer were depolarized for 1 h in 10 μM CCCP. Cells were fixed and imaged. Pacer puncta were counted using an automated particle counting program. 20 cells were analyzed and averaged in each biological replicate, and each cell's puncta count was normalized to the average untreated puncta count for that day. Three to four of these experiments were performed and averaged. Colored dots indicate technical replicates, black dots indicate averages for each experimental day, black horizontal line indicates averages of biological replicates. P value calculated using a two-tailed $t$ test of biological replicates. **(C)** Fold change in total autophagosome area. Cells imaged in A were masked and thresholded, and the total LC3 positive area was quantified. The total area for each cell was normalized to the average area for the mCherry transfected condition, and these values were averaged to calculate the value for a single biological replicate. This process was repeated in four independent experiments. P values were determined using Tukey's multiple comparison test. **(D)** Fold change in LC3+/LAMP1+ positive area, determined as in C. **(E)** Fraction of total LC3 area that is also positive for LAMP1 as a metric for autophagosome-lysosome fusion efficiency, analysis performed as in C.

protein kinases, TBK1 and LRRK1. TBK1 is well-established as a central player in PINK1/Parkin mitophagy (Heo et al., 2015; Moore and Holzbaur, 2016; Richter et al., 2016), and a mitophagy role for LRRK1 was more recently reported (Fujita et al., 2022).

The best-explored function of RAB phosphorylation in autophagy is the regulation of JIP4-dependent transport of mature autophagosomes in axons by LRRK2 phosphorylation of RAB29 (Boecker et al., 2021). Here, we have discovered that RAB

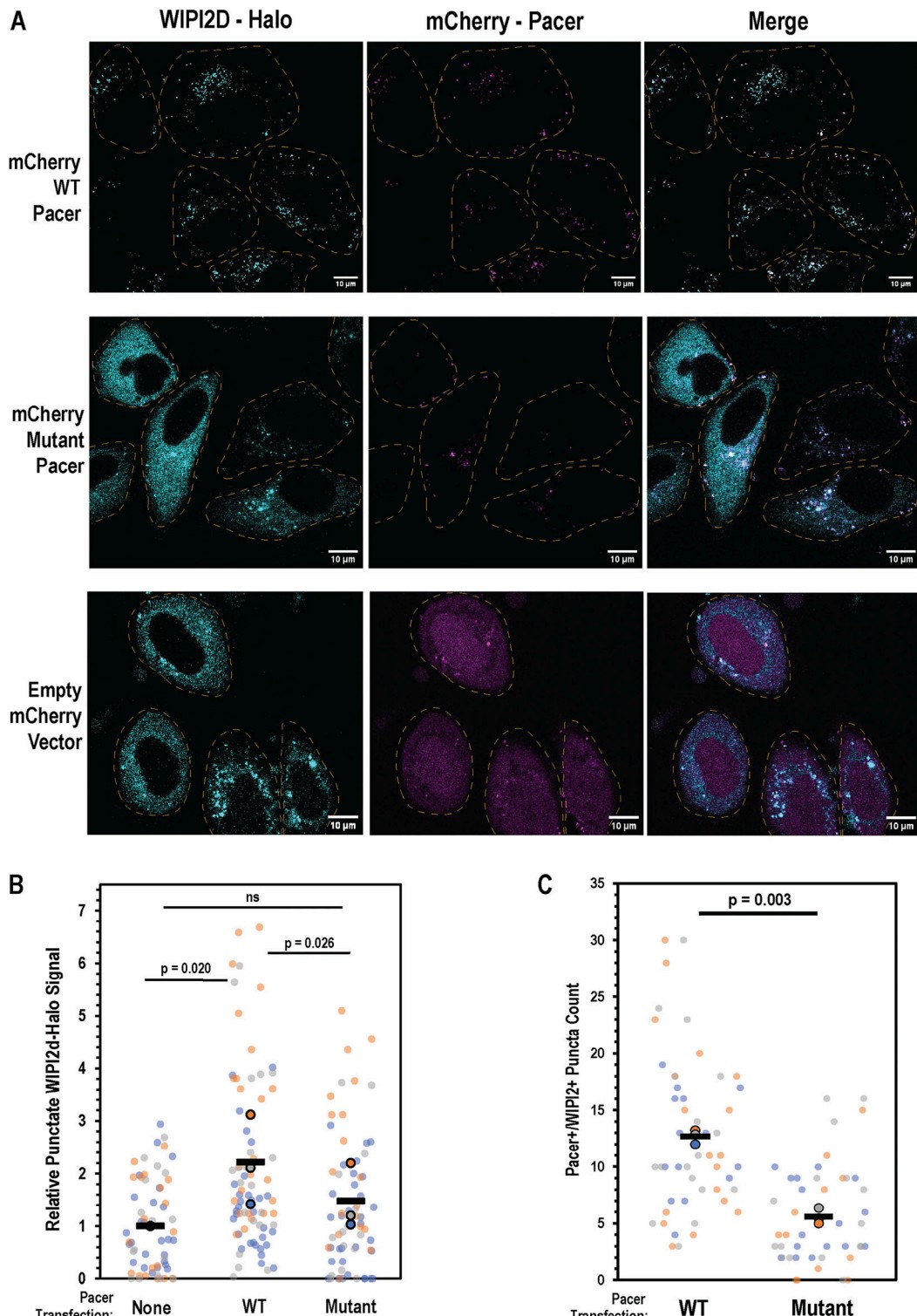

Figure 7. **Pacer expression promotes the formation of WIPI2 puncta upon depolarization. (A)** Pacer KO HeLa cells expressing Parkin and either wild-type Pacer, mutant PacerK534N, R623T, or empty vector were depolarized for 1 h using 10 µM Oligomycin A and 5 µM Antimycin A following transient transfection with HaloTag-WIPI2B. **(B)** For each independent trial, ~20 cells were imaged and the total above-cytosol background, or punctate, WIPI2 signal was measured. Three trials were performed, and the trials performed in parallel are indicated by colors. Trial averages are indicated by bolded dots, and horizontal bars indicate averages across all three trials. P value is the result of a one-tailed, paired $t$ test between the trial averages in each category. **(C)** Quantification of the number of puncta positive for both Pacer and WIPI2 on a per cell basis. Replicates are indicated as in B, and P value is also the result of a one-tailed, paired $t$ test.

phosphorylation can also regulate the process of autophagosome biogenesis itself.

It was previously reported that pSer72 RAB7A preferentially recruits the FLCN:FNIP1 complex relative to unphosphorylated RAB7A (Heo et al., 2018). FLCN:FNIP1/2 is a GTPase-activating protein complex (GAP) for the RagC/D GTPases involved in activating mTORC1 to phosphorylate and inactivate the lysosomal and autophagic transcriptional regulator TFEB (Cui et al., 2023; Napolitano et al., 2020; Tsun et al., 2013). The FNIP1-containing FLCN:FNIP isocomplex is centrally involved in sensing and responding to changes in mitochondrial metabolism (Manford et al., 2021; Malik et al., 2023). FLCN KO does not phenocopy the effect of pRAB7A phosphorylation on the recruitment of ATG9A to mitochondria (Heo et al., 2018), a key step in mitophagy initiation. Thus, it was postulated that RAB7A phosphorylation must have additional effects on the autophagy machinery. This concept is consistent with our own finding that purified FLCN:FNIP1 binds robustly to dephosphorylated RAB7A in vitro (unpublished data). While the possibility of yet other effects of RAB7A on the mitophagy machinery cannot be excluded, RAB7A Ser72 phosphoregulation of Rubicon and Pacer is, in principle, sufficient to account for the pRAB7A dependence of PINK1/Parkin mitophagy.

Most models for the regulation of selective autophagy focus on the first step: substrate recognition by cargo receptors (Stolz et al., 2014; Adriaenssens et al., 2022). The assumption is often made that once initiated, the downstream cascade of protein interactions in autophagosome biogenesis is the same in bulk autophagy and the various forms of selective autophagy. Recent evidence shows that the situation is more complicated. In OPTN-dependent mitophagy, for example, the order of recruitment of the core autophagy components ULK1 and PI3KC3-C1 is inverted compared with most other forms of autophagy (Nguyen et al., 2023). Now we have uncovered a PINK1/Parkin mitophagy-specific mode of regulation of PI3KC3-C2 via Rubicon and Pacer, suggesting that selective autophagy subtype-specific regulator mechanisms may permeate the entire process of autophagosome biogenesis. The existence of these mechanisms may provide new opportunities for the modulation of specific types of autophagy for therapeutic benefit.

## Materials and methods

### Structural modeling
To generate a homology model for the Pacer RH domain, the Pacer RH sequence (corresponding to residues 412 through 654) in FASTA format was downloaded from Uniprot (RRID: SCR_002380) (accession date: September 1, 2021) and uploaded to the Biozentrum Swiss-Model (RRID:SCR_018123). This sequence was aligned against the Rubicon sequence and a model was generated using a published crystal structure of the Rubicon RH domain in complex with RAB7A (PDB: 6wcw [Bhargava et al., 2020]).

### Plasmid construction
To generate MBP-tagged RH domains for purification and subsequent in vitro binding experiments, the RH domain-encoding sequences of Rubicon and Pacer were amplified via PCR, purified via gel electrophoresis using a 1% agarose gel, and then subcloned into pET His10 MBP Asn10 TEV LIC cloning vector (2CT-10) via ligation independent cloning (LIC). XL10 Ultra-competent E. coli (Agilent Technologies) were transformed with this plasmid and then plated on ampicillin selective agar. Individual colonies were picked via a sterile loop and grown overnight in 5 ml of LB supplemented with ampicillin. Purified plasmid DNA was isolated via a commercial spin miniprep kit (Qiagen), and cloning was confirmed via Sanger sequencing.

To generate point mutants of the Pacer RH domain, "round-the-horn site-directed mutagenesis" was employed. Briefly, non-overlapping primers, adjoining each other and "pointing in the opposite direction" were designed. One of the primers contained the desired mutations and the primers were phosphorylated using polynucleotide kinase (NEB) prior to PCR such that each PCR product strand could be immediately ligated to itself. Following PCR using Q5 polymerase (NEB), the product was gel-purified on a 1% agarose gel and ligated via incubation with T4 DNA ligase (NEB) prior to transformation into XL10 E. coli. The N821K, T911R, and K916R mutants of the Rubicon RH domain were synthesized by Genscript in a pMAL-c5E vector.

To generate mCherry-tagged WT Rubicon and Pacer for use in confocal imaging and autophagy flux experiments, primers were designed to amplify the sequence encoding full-length Rubicon and Pacer, respectively, using existing Rubicon and Pacer plasmids as templates (Chang et al., 2019) while incorporating a BglII restriction site upstream of the open reading frame and a stop codon followed by a SalI restriction site downstream of the open reading frame. Following PCR amplification, the product, as well as a stock of pmCherryC1 mammalian expression vector (Clontech) was digested using BglII and SalI (NEB) according to the manufacturer's instructions. The digested PCR product and vector were purified via electrophoresis on a 1 and 0.5% agarose gel, respectively. Subsequently, the Rubicon/Pacer full-length sequence was ligated into the pmCherry-C1 vector via an overnight incubation with T4 DNA ligase (NEB) at 16°C.

RH domain mutants of full-length Pacer were generated via Gibson Assembly. Briefly, a fragment consisting of Pacer residues 1–412 was amplified via PCR, with a 20-bp overlap with the pmCherryC1 sequence upstream of the amplified sequence via primer and a 20 bp overlap with the RH domain included downstream of the fragment. Simultaneously, the mutant RH domains in the 2CT-10 cloning vector were amplified via PCR, with an additional 20-bp overlap with pmCherryC1 included downstream of the Pacer RH sequence and a 20-bp overlap with the Pacer 1–412 transcript included upstream. pmCherryC1 was linearized via incubation with BglII and SalI, and then the linearized vector, Pacer fragment 1–412 PCR product, and mutant Pacer RH domain PCR product were mixed with Gibson Assembly master mix (NEB) in a 1:3:3 M ratio prior to incubation at 50°C for 15 min and transformation into XL10 E. coli. Plasmids were subsequently purified via overnight culture followed by purification via a commercial endotoxin-free spin midi prep kit (Qiagen).

To prepare for stable cell-line generation, mCherry-tagged full-length Pacer and the Pacer pRAB7A–binding

mutant were subcloned in a pCDH lentiviral vector. Briefly, both the pCHD1 vector and pmCherry-RH protein mammalian expression plasmids were digested using NheI and BamHI before being purified via gel electrophoresis. Bands corresponding to mCherry—RH protein were isolated and purified before being ligated into the pCDH backbone via T4 DNA polymerase. Subsequently, the final pCDH—RH protein construct was transformed into One Shot Stbl3 Chemically Competent *E. coli* (Thermo Fisher Scientific) before being selected via an antibiotic and grown in an overnight culture. A maxiprep was performed to purify the plasmid (Qiagen), and the plasmid was subsequently used for stable cell line generation via lentiviral transduction.

*E. coli* expressible constructs were verified via sequencing and SDS-PAGE of target proteins. Mammalian expressible constructs were verified via sequencing, visibility of fluorescent tag, and Western blot against construct protein in a knockout background.

### Protein expression and purification
Plasmids were transformed into BL21 star (DE3) *E. coli* (Agilent technologies) and individual colonies were picked. For purification, 20 ml of overnight culture was inoculated and then diluted into 4 liters of lysogeny broth supplemented with maintenance antibiotics. When purifying MBP-Rubicon RH or MBP-Pacer RH, cultures were also supplemented with 150 μM ZnCl$_2$. Cultures were grown at 37°C until OD600 = 0.55 was reached and then cultures were chilled in ice water baths for 15 min before being induced with 350 μM isopropyl-β-D-thiogalactoside (IPTG). Induced cultures expressed protein at 18°C for 16–20 h before being harvested via centrifugation. Harvested cells were resuspended in a buffer consisting of 50 mM HEPES pH 7.5, 150 mM NaCl, 2 mM MgCl2, and 10 mM TCEP, and a tablet of cOmplete protease inhibitor cocktail (Roche). Cells were lysed via sonication, and the lysate was clarified by centrifugation at 34,500 *g* for 1 h at 4°C.

MBP-tagged RH domain was isolated via passing cleared lysate over a gravity amylose resin column containing 5 ml of settled resin (NEB). The column was washed with 150 ml of buffer before eluting protein with wash buffer supplemented with 20 mM maltose. As a final polishing step, the protein was concentrated down to 500 μl before being loaded onto a Superdex 75 Increase 10/300 Gl size-exclusion column (Cytiva). Fractions were analyzed via SDS-PAGE, and pure fractions were pooled and concentrated before being snap-frozen in liquid nitrogen.

GST-tagged RAB7A was similarly isolated via passing cleared lysate over a gravity column packed with 5 ml of Glutathione Sepharose 4B (GE Healthcare), followed by elution using 10 mM reduced glutathione dissolved in 50 mM Tris-HCl pH 8.0. To cleave the GST tag, purified TEV protease was added in a 1:20 protease: RAB7A ratio by mass, and the solution was dialyzed overnight into a 50 mM HEPES 7.5, 150 mM NaCl, 2 mM MgCl2, 10 mM TCEP buffer. Protease and cleaved GST were removed from the solution by passing the dialyzed and cleaved protein solution over a 10 ml Hispur Ni-NTA gravity column (Thermo Fisher Scientific) three times.

Subsequently, the cleaved RAB7A was concentrated down to 500 μl and further purified via size exclusion as with the purified RH domains.

### Bead binding
A 40 μl solution of 25% (vol/vol) amylose resin (NEB) was co-incubated with 500 nM MBP-RH domain and 4 μM Alexa Fluor 647-labeled RAB7A (DOL = 0.9; Life Technologies) in a buffer consisting of 50 mM HEPES 7.5, 150 mM NaCl, 2 mM MgCl2, and 10 mM TCEP. Samples were incubated on a rocker at room temperature for 1 h. The resin was collected using a tabletop centrifuge, the supernatant was aspirated off, and resin was washed three times with 1 ml of room temperature buffer. Beads were resuspended in 200 μl buffer and transferred to an eight-well confocal imaging chamber (LabTek). Images were collected on a Ti2 Nikon confocal microscope with a 63x Plan Apochromat 1.4 NA objective, and a ti2 eclipse camera using Nis elements capture software (RRID: SCR_014329). Binding was quantified via the Alexa Fluor 647 channel. Bead fluorescence intensity was assessed in ImageJ (RRID:SCR_003070) by selecting an in-focus bead, defining an ROI that includes the bead edge, and recording the maximum pixel intensity. Any changes in image brightness performed for figure legibility were applied uniformly across images in a figure.

### Phosphorylation reactions
Purified RAB7A was phosphorylated via incubation with purified strep-tagged TBK1. Briefly, a solution consisting of 4 μM RAB7A, 0.4 μM TBK1, 50 mM HEPES 7.5, 150 mM NaCl, 10 mM MgCl2, and 200 μM ATP was incubated at room temperature for 3 h. Strep-tagged TBK1 was removed from the solution via incubation with Strep-Tactin Sepharose (Cytiva Life Sciences). The resin was collected via a filtered gravity column yielding pure, phosphorylated RAB7A.

To assess the degree of phosphorylation, a 15-μg pRAB7A sample was then buffer-exchanged into MQ water via dilution and concentration in a centrifugal concentrator, and the samples were subsequently analyzed via electrophoresis on a SuperSep Phos-tag Precast gel (Wako Fujifilm). Complete band shift of the RAB7A band post phosphorylation confirms stoichiometric phosphorylation of the protein.

### Isothermal titration calorimetry
To prepare proteins for isothermal titration calorimetry (ITC) binding studies, recombinantly purified RAB7A and MBP-Rubicon RH were co-dialyzed overnight at 4°C into a degassed buffer containing 50 mM HEPES 7.5, 150 mM NaCl, 2 mM MgCl$_2$, and 10 mM TCEP. The next morning, RAB7A was concentrated to 800 μM and MBP-Rubicon RH to 80 μM using a centrifugal concentrator. The concentrated protein solutions were again degassed via exposure to vacuum, allowed to come to room temperature, and loaded onto a Microcal PEAQ-ITC (Malvern Panalytical), with the MBP-Rubicon RH contained in the ITC chamber and the RAB7A solution in the syringe. The titration was conducted, and the resulting binding curve was fit using the PEAQ-ITC's native curve-fitting and analysis software.

## Tissue culture

All cell lines were grown and maintained in Dulbecco's Modified Eagle Medium (Corning) supplemented with 10% fetal bovine serum (Gibco) and 1x Penicillin-Streptomycin (Thermo Fisher Scientific). To depolarize cells for mitophagy induction, the growth medium was supplemented with 10 µM Oligomycin and 5 µM Antimycin A (Sigma-Aldrich), or with 10 µM carbonyl cyanide m-chlorophenyl hydrazone (CCCP) (Medchem Express). To induce starvation autophagy, cells were washed into Earle's Balanced Salt Solution (EBSS) (Sigma-Aldrich).

## Generation of knockout lines using CRISPR/Cas9 gene editing

RUBICON and PACER KO lines in this study were generated using CRISPR/Cas9 gene editing. CRISPR guide RNAs (gRNAs) targeting a common exon of splicing variants of each gene of interest were cloned into BBsI-linearized pSpCas9(BB)-2A-GFP vector (#48138; Addgene) using NEBuilder HiFi DNA Assembly (New England Biolabs) as described in the following protocol: https://doi.org/10.17504/protocols.io.j8nlkkzo6l5r/v1 (Nguyen, 2023). gRNA constructs were transfected into HeLa S3 cells (https://www.atcc.org/products/ccl-2.2) with X-tremeGENE 9 (Roche) overnight. With fluorescence-activated cell sorting (FACS), GFP-positive single cells were sorted into individual wells of 96-well plates. To identify knockout cell lines, single-cell colonies were expanded and initially screened via Western blot. In the absence of an antibody, the PACER KO line was confirmed via sequencing alone. The CRISPR-targeted regions of the knockout clones and wild-type cells were amplified via PCR and sequenced using primers that annealed to the amplified product. Synthego ICE v2 CRISPR Analysis Tool (https://synthego.com/products/bioinformatics/crispr-analysis) (RRID: SCR_024508) was used to compare the sequencing data of the control and knockout cell lines to confirm the presence of indels.

## Generation of stable cell lines

To prepare the retrovirus transduction solution, HEK293T cells were transfected for 6 h with pMRX-IP-based or, pMRX-IB-based retroviral plasmids containing HaloTag autophagy reporters or HA-tagged Parkin, pCMVR8.74, and pCG-VSV-G using TransIT-LT-1 transfection reagent (Mirus). Subsequently, the medium was replaced with DMEM and the HEK cells were allowed to produce pseudotyped retrovirus for 3 days. The medium was harvested, cleared via centrifugation, and concentrated 10-fold via Lenti-X Concentrator solution (Takara). The retroviral solution was added to WT, Pacer knockout, and Rubicon knockout HeLa cells in a 12-well plate at 50% confluence. Cells were left to incubate for 24 h before swapping medium to fresh DMEM and assessing transfection efficiency via fluorescence. Transfected cells were selected via incubation with puromycin or blasticidin depending on the transduction plasmid, and uniform expression of autophagy reporters across cell types was confirmed via Western blot and fluorescence microscopy.

## Immunofluorescence imaging and colocalization

Cells were seeded 2 days in advance on chambered glass slides treated with poly-L-lysine. Cells were allowed to grow to 80% confluence and then transfected with mCherry-labeled RH proteins using lipofectamine 3000 (Thermo Fisher Scientific). Cells were allowed to recover for 48 h before being depolarized using CCCP or Oligomycin A/Antimycin A for 1 h or starved via incubation in EBSS, then washed with PBS and fixed with 4% paraformaldehyde for 15 min. Cells were permeabilized via a 15-minute incubation in Streptolysin O from *Streptococcus pyogenes* (Sigma-Aldrich), washed three times with PBS, and then blocked via a 30-min incubation in 2% BSA dissolved in PBS. Cells were washed and incubated for 1 H in a 1:200 primary solution of Rab7 (ab137029; Abcam) or pRAB7A primary antibody (ab302494; Abcam). The pRAB7A antibody was previously demonstrated to be specific to pS72 RAB7A (Malik et al., 2021). Cells were washed three times with PBS and then incubated in a 1:1,000 solution of anti-Rabbit secondary antibody conjugated to Alexa 488 for 1 h (A-11034; Thermo Fisher Scientific). Cells were again washed three times and imaged immediately. Images were captured using the same setup as in the bead-binding experiments.

To analyze and quantify colocalization, images were opened in FIJI (RRID:SCR_002285), an open-source distribution of ImageJ. To subtract the background of each channel, a portion of the image expected to be dark, such as a nucleus, was defined, and the average signal intensity was measured. This average signal intensity was subtracted from the image as a whole using FIJI's *Math > Subtract* function. Subsequently, lasso ROIs are used to define individual cell boundaries, and the Mander's colocalization coefficient with a Costes' threshold regression was calculated using the *BIOP JaCoP* plugin (version 2.11, https://github.com/radsz/jacop, https://doi.org/10.1111/j.1365-2818.2006.01706.x [Bolte and Cordelières, 2006], RRID: SCR_025164). When images are brightened for clarity, gamma adjustments are made uniformly across compared images.

For the LC3 puncta formation assays, Pacer KO+ HA-Parkin HeLa cells were transfected with Halo-LC3 and either mCherry, mCherry-PacerWT, or PacerK534N, and R623T using Lipofectamine 2000 (Cat#11668030; Invitrogen). After 18–24 h of transfection, cells were treated with Ethanol and DMSO or Antimycin A and Oligomycin A for 4 h. During the last hour of treatment, cells were labeled with Janelia Fluor 646-Halo ligand (Cat#GA1120; Promega) for 15 min followed by a wash and incubation in fresh treatment media for another 45 min. Cells were then fixed and permeabilized with ice-cold methanol at −20°C for 8 min. After this, cells were blocked at RT in blocking buffer (5% goat serum and 1% BSA in 1X Phosphate Buffer Saline [PBS] buffer) for 90 min. Following blocking, cells were incubated overnight at 40°C with primary antibodies diluted in blocking buffer (HSP60: Cat#SAB4501464 IF: 1:100; Rabbit polyclonal sigma-Aldrich; mCherry: Cat#ab205402, IF: 1:200). After 14–16 h of incubation, cells were washed thrice with 1X PBS. Cells were then incubated for 1 h at RT with a blocking buffer containing the corresponding secondary antibody (Alexa Fluor 488 goat anti-rabbit IgG [H + L] Cat#A11034; Invitrogen; Alexa Fluor 546 goat anti-chicken IgG [H + L] Cat#A11040; dilution: 1:1,000; Invitrogen) and Janelia Fluor 646-Halo ligand. Samples were washed thrice in 1XPBS and then mounted in Prolong Gold (P36930; Life Technologies). Images were acquired on a Perkin-Elmer UltraView Vox spinning disk confocal on a Nikon Eclipse Ti

Microscope with an Apochromat 100x 1.49 N.A. oil-immersion objective and a Hamamatsu CMOS ORCA Fusion (C11440) camera with VisiView (Visitron).

## HaloTag processing assay

Cells stably expressing LC3-Halo or Su9-Halo were seeded for 80% confluence on the day of the assay in 12-well tissue culture plates. To pulse-label the reporter, cells were incubated for 20 min in 100 nM cell-permeable Halotag conjugated to Janelia Fluor 549 dye (Promega) before washing twice with PBS. Cells were incubated in a growth medium or in a growth medium supplemented with 10 µM oligomycin and 5 µM antimycin, or in a growth medium supplemented with 1 mM deferiprone (Millipore Sigma), or in Earle's Balanced Salt Solution before harvesting cells via scraping. Incubation times are listed in the respective figure caption. Cells were lysed and the protein concentration was quantified in a plate reader using a Pierce BCA protein quantification kit (Thermo Fisher Scientific). For each condition, 20 µg of protein was loaded onto a gel for SDS-PAGE and then the gels were imaged using a ChemiDoc fluorescent imager (Biorad).

Images of gels (Fig. S4) were imported into FIJI and analyzed using the *Gel Analyzer* tool. Briefly, selections were defined that contained the bands of interest, and the integrated signal for each band was recorded. The intensity of the processed HaloTag band as a fraction of the total Halo signal was calculated.

## Western blotting

Adherent cells to be analyzed were washed twice with PBS and then scraped into a microcentrifuge tube. Cells were pelleted and lysed, and their concentration was measured as performed for the HaloTag processing assay. 20 µg of protein was loaded into each lane. Protein was transferred to a nitrocellulose membrane (GE Healthcare) using a TransBlot Turbo transfer device (Biorad) and the device's default protocol. After transfer, the membrane was cut lengthwise into strips to isolate bands of particular molecular weights before being washed with TBST and incubated at room temperature for 1 h in a 5% blocking buffer solution (Biorad). The Western blot strips were then incubated overnight at 4°C in a solution consisting of the primary antibody diluted into a blocking buffer. The next morning, the Western blot strips were washed in TBST and incubated in a solution of secondary antibodies diluted in a blocking buffer for 1 h at room temperature. The strips were rigorously washed and prepared for chemiluminescent imaging via incubation in SuperSignal West Femto ECL detection substrate (Thermo Fisher Scientific) for 5 min prior to imaging on a ChemiDoc (BioRad).

## Autophagosome fluorescence microscopy live imaging

For live-cell imaging, Pacer^KO+HA-Parkin HeLa cells were first co-transfected with pCIG2-LAMP1-mNeon Green and pHaloTag-LC3B and either pmCherry, pmCherry-Pacer^WT, or pmCherry-Pacer^K534N/R623T. After 16–18 h of transfection, cells were treated with Antimycin A and Oligomycin A for 4 h. For imaging, the culture media was replaced with Leibovitz's L-15 medium (11415064; Gibco) supplemented with 10% fetal bovine serum and 1% Glutamax along with Antimycin A and Oligomycin A. Imaging was performed on a PerkinElmer UltraView Vox spinning disk confocal on a Nikon Eclipse Ti Microscope with an Apochromat 100x 1.49 N.A. oil-immersion objective and a Hamamatsu CMOS ORCA Fusion (C11440-20UP) camera with VisiView (Visitron). mCherry was expressed to a higher level than either pmCherry-Pacer^WT or pmCherry-Pacer^K534N/R623T, resulting in higher fluorescent intensities per cell. Thus, the red channel for mCherry-expressing cells was acquired at a lower laser power and shorter exposure time than those used for both mCherry-Pacer^WT and mCherry-Pacer^K534N/R623T expressing cells; however, the LAMP1 and LC3B channels were imaged at the same laser power and exposure time across all three constructs. Imaging parameters were kept constant across biological replicates. Quantification was performed on individual slices from a Z-stack. Images from the LC3 and LAMP1 channels were thresholded and binarized using Yen thresholding in ImageJ. To quantify the area of colocalization, the "AND" function from Image Calculator was used. The area of colocalization was then quantified using Analyze Particles.

## WIPI2 puncta formation assay

Hela cells expressing Parkin were seeded 2 days in advance on chambered glass slides treated with poly-L-lysine. Cells were allowed to grow to 80% confluence and then transfected with HaloTag-WIPI2B and mCherry-Pacer using lipofectamine 3000 (Thermo Fisher Scientific). Cells were allowed to recover for 48 h before being depolarized using Oligomycin A/Antimycin A for 45 min. After this incubation, cells were labeled using 1 µM cell-permeable Oregon Green Halo Ligand (Promega) diluted in a depolarization medium. Cells were incubated for an additional 15 min, washed twice with PBS, and immediately fixed with 4% paraformaldehyde for 15 min. Cells were immediately imaged on a confocal microscope.

To analyze and quantify WIPI2 puncta formation, images were opened in FIJI, an open-source distribution of ImageJ. To subtract the background of each channel, a portion of the image containing only the cytosolic WIPI2 signal was defined and the average signal intensity was measured. This average signal intensity is subtracted from the image as a whole using FIJI's math > subtract function. Subsequently, lasso ROIs were used to define individual cell boundaries and the measure function was used to calculate the total above-background, punctate WIPI2 signal.

To measure colocalization, ROIs were defined around cells and then images were processed using FIJI's auto threshold > max entropy automatic thresholding function. This is performed on both WIPI2 and Pacer channels. Then, images were multiplied together using FIJI's image calculator > multiply function. The resulting image contains only pixels positive for both WIPI2 and Pacer. Then, analyze > analyze particles was used on each ROI with a size of 0-infinity and a circularity of 0–1 to automatically count the number of particles. Key resources are shown in Table 1.

| Reagent | Source | Identifier |
|---|---|---|
| pET His10 MBP Asn10 TEV LIC cloning vector (2CT-10) | Addgene | RRID: Addgene_55209 |
| XL10 Ultracompetent *E. coli* | Agilent Technologies | Cat: 200314 |
| pMAL-c5E | Addgene | RRID: Addgene_161783 |
| pmCherryC1 | Clontech | Cat: 632524, ID: PT3975-5 |
| pCDH | SBI | Cat: CD500B-1, RRID: Addgene_72265 |
| BL21 star (DE3) *E. coli* | Agilent Technologies | Cat: C601003 |
| HeLa S3 cells | ATCC | ATCC Cat: CCL-2.2, RRID: CVCL_0058 |
| HEK 293T | UC Berkeley Cell Culture Facility | RRID: CVCL_0063 |
| pSpCas9(BB)-2A-GFP vector | Addgene | RRID: Addgene_48138 |
| pMRX-IP-LC3-Halo | Addgene | RRID: Addgene_184899 |
| pMRX-IP-Parkin | Addgene | RRID: Addgene_38248 |
| pMRX-IB-Su9-GFP-Halo | Addgene | RRID: Addgene_184905 |
| pmCherryC1-Rubicon | Addgene | RRID: Addgene_216697 |
| pmCherryC1-Pacer | Addgene | RRID: Addgene_216698 |
| pmCherryC1-Pacer K534N, R623T | Addgene | RRID: Addgene_216699 |
| 2CT-PacerRH | Addgene | RRID: Addgene_216700 |
| 2CT-PacerRH K534N | Addgene | RRID: Addgene_216701 |
| 2CT-PacerRH R623T | Addgene | RRID: Addgene_216702 |
| 2CT-PacerRH R628K | Addgene | RRID: Addgene_216703 |
| 2CT-PacerRH K534N, R623T | Addgene | RRID: Addgene_216704 |
| 2CT-PacerRH K534N,R623T,R628K | Addgene | RRID: Addgene_216705 |
| pMAL-c5e-Rubicon RH | Addgene | RRID: Addgene_216706 |
| pMAL-c5e-Rubicon RH N821K, T911R, K916R | Addgene | RRID: Addgene_216707 |
| 2GT-RAB7A Q67L | Addgene | RRID: Addgene_216708 |
| pCAG-TSF-TBK1 | Addgene | RRID: Addgene_216709 |
| pCG-VSV-G | Addgene | RRID: Addgene_8454 |
| pCMVR8.74 | Addgene | RRID: Addgene_22036 |
| Halo-WIPI2B | Addgene | RRID: Addgene_175025 |

| Reagent | Source | Identifier |
|---|---|---|
| pCIG2-IRES-Lamp1-mNeon Green | Addgene | RRID: Addgene_219441 |
| pHTN-LC3B | Addgene | RRID: Addgene_219440 |
| pS72 Rab7 rabbit monoclonal | Abcam | Cat: ab302494, RRID: AB_2933985 |
| Anti Rab7 rabbit monoclonal antibody | Abcam | Cat: ab137029, RRID: AB_2629474 |
| Anti Pacer rabbit antibody | Millipore Sigma | Cat: HPA026614, RRID: AB_10602693 |
| Rubicon (D9F7) rabbit monoclonal antibody | Cell Signaling | Cat: #8465, RRID: AB_10891617 |
| Anti-alpha tubulin antibody [DM1A]—Loading Control | Abcam | Cat: ab7291, RRID: AB_2241126 |
| Goat anti-rabbit IgG (H + L) highly Cross-adsorbed secondary antibody, Alexa Fluor 488 | Thermo Fisher Scientific | Cat: A11034, RRID: AB_2576217 |
| Goat anti-rabbit IgG H&L (HRP) | Abcam | Cat: ab6721, RRID: AB_955447 |
| HSP60 rabbit polyclonal antibody | Sigma-Aldrich | Cat# SAB4501464, RRID: AB_10746162 |
| Anti mCherry chicken polyclonal antibody | Abcam | Cat: ab205402, RRID: AB_2722769 |
| HeLa S3 RUBCN KO clone 15 | This paper | AC: CVCL_D5AG |
| S3 HeLas RUBCN KO + HA-Parkin/pSu9-HaloTag7-mGFP | This paper | AC: CVCL_D5AJ |
| HeLa S3 RUBCN KO/HaloTag7-LC3 | This paper | AC: CVCL_D5AK |
| HeLa S3 RUBCNL KO clone 17 (Pacer KO) | This paper | AC: CVCL_D5AH |
| HeLa S3 RUBCNL KO (Pacer KO)/ HA-PRKN/pSu9-HaloTag7-mGFP | This paper | AC: CVCL_D5AL |
| HeLa S3 RUBCNL KO (Pacer KO)/ HaloTag7-LC3 | This paper | AC: CVCL_D5AM |
| HeLa S3 PRKN | This paper | AC: CVCL_D5AN |
| HeLa S3 HA-PRKN/HaloTag7-mGFP | This paper | AC: CVCL_D5AP |
| HeLa S3 HaloTag7-LC3 | This paper | AC: CVCL_D5AQ |

## Statistical analysis

All statistical analyses were performed as indicated in the figure captions. Analysis was performed in either Microsoft Excel (RRID:SCR_016137) or Graphpad Prism (RRID:SCR_002798). Data distribution was assumed to be normal but was not formally tested.

## Online supplemental material

The supplementary material consists of two tables and four figures. Fig. S1 shows the validation of CRISPR KO of Rubicon and Pacer. Fig. S2 shows the pS72 Rab7 formation following depolarization, starvation, and iron chelation. Fig.

S3 shows the immunofluorescence of mitochondria alongside LC3B. Fig. S4 shows the MW verification of HaloTag flux reporters. Table S1 shows the oligonucleotides for CRISPR KO cell lines. Table S2 shows the details of CRISPR sequences and the genotyping results of knockout cell lines in this study.

### Data availability
Protocols for the methods used are available online (DOI: dx.doi.org/10.17504/protocols.io.3byl4qy7jvo5/v1). Protocol for live cell imaging is available online (DOI: dx.doi.org/10.17504/protocols.io.5qpvoknobl4o/v1). Protocol for CRISPR KO cell line generation is available online (https://doi.org/10.17504/protocols.io.j8nlkkzo6l5r/v1). Datasets produced are available online on Zenodo, organized by figure: Fig. 1 (https://doi.org/10.5281/zenodo.10046037), Fig. 2 (https://doi.org/10.5281/zenodo.10896125), Fig. 3 (https://doi.org/10.5281/zenodo.10896130), Fig. 4 (https://doi.org/10.5281/zenodo.10896135), Fig. 5 (https://doi.org/10.5281/zenodo.10896142), Fig. 6 (https://doi.org/10.5281/zenodo.10482934), Fig. 7 (https://doi.org/10.5281/zenodo.10896364), Fig. S2 (https://doi.org/10.5281/zenodo.10896376), Fig. S3 (https://doi.org/10.5281/zenodo.10798586), Fig. S4 (https://doi.org/10.5281/zenodo.10791411). Source data for gels and Western blots are available online on Zenodo (https://doi.org/10.5281/zenodo.10798584).

## Acknowledgments
This work was supported by the Michael J. Fox Foundation for Parkinson's Research (MJFF) and Aligning Science Across Parkinson's (ASAP) initiative. MJFF administers the grant ASAP-000350 on behalf of ASAP and itself. (J.H. Hurley, E.L.F. Holzbaur, and M. Lazarou.).

Author contributions: D.A. Tudorica: Conceptualization, Data curation, Formal analysis, Investigation, Methodology, Validation, Visualization, Writing—original draft, Writing—review and editing; B. Basak: Formal analysis, Investigation, Methodology, Visualization, Writing—review and editing; A.S. Puerta Cordova: Formal analysis, Investigation, Methodology; G. Khuu: Resources, Visualization; K. Rose: Methodology, Resources; M. Lazarou: Funding acquisition, Project administration, Resources, Supervision, Writing—review and editing; E.L.F. Holzbaur: Conceptualization, Data curation, Funding acquisition, Project administration, Resources, Supervision, Writing—review and editing; J.H. Hurley: Conceptualization, Funding acquisition, Project administration, Supervision, Writing—original draft, Writing—review and editing.

Disclosures: All authors have completed and submitted the ICMJE Form for Disclosure of Potential Conflicts of Interest. M. Lazarou reported other from Automera outside the submitted work. J. Hurley reported grants from Casma Therapeutics during the conduct of the study; personal fees from Casma Therapeutics outside the submitted work; and "Unrelated work in the lab is supported by Genentech and Hoffmann-La Roche." No other disclosures were reported.

Submitted: 5 September 2023

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

# Supplemental material

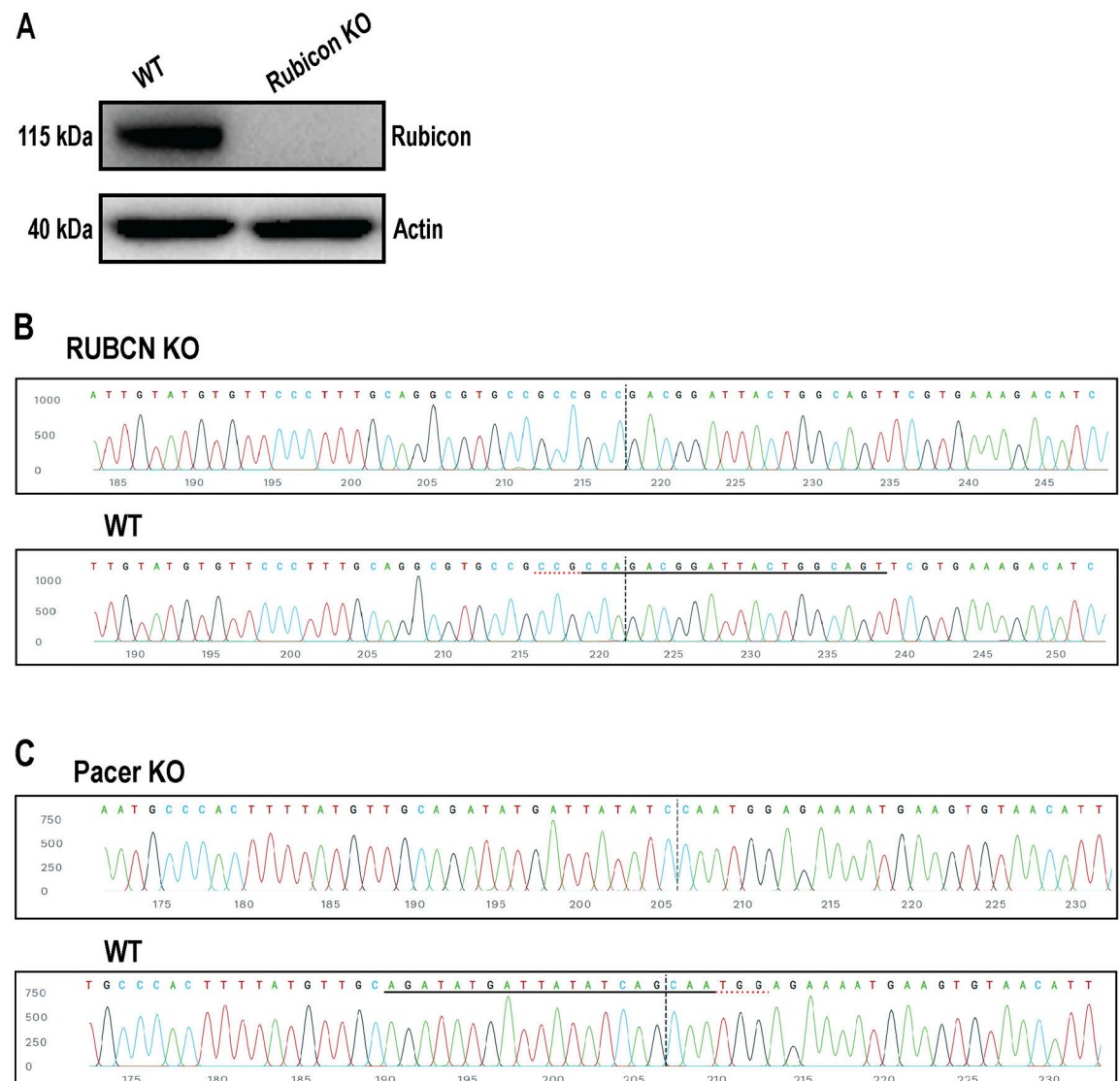

Figure S1. **Validation of CRISPR KO of Rubicon and Pacer. (A)** Immunoblot confirming knockout of Rubicon. Rubicon blot with molecular weight marker is depicted in Fig. S4 D. **(B)** Sanger sequence view showing Rubicon KO and WT sequences in the region around the guide sequence. The horizontal black line indicates the guide sequence. The horizontal red line is the PAM site. The vertical black dotted line represents the actual cut site. **(C)** Sequence view showing Pacer KO represented as in B. Source data are available for this figure: SourceData FS1.

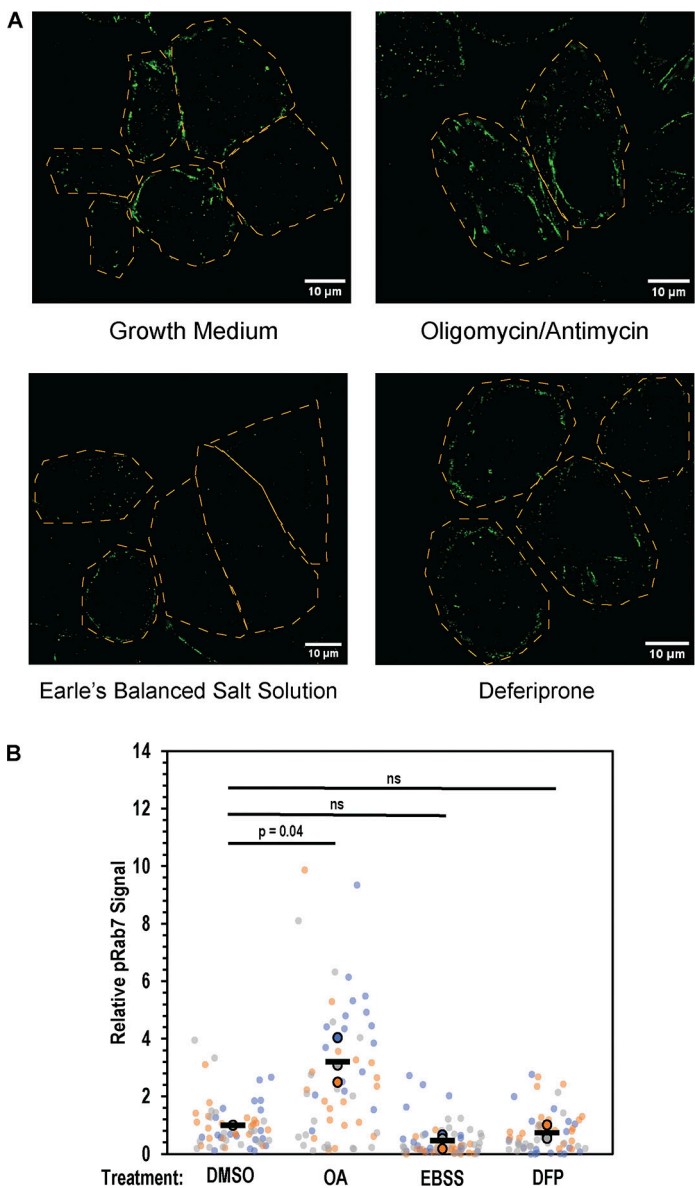

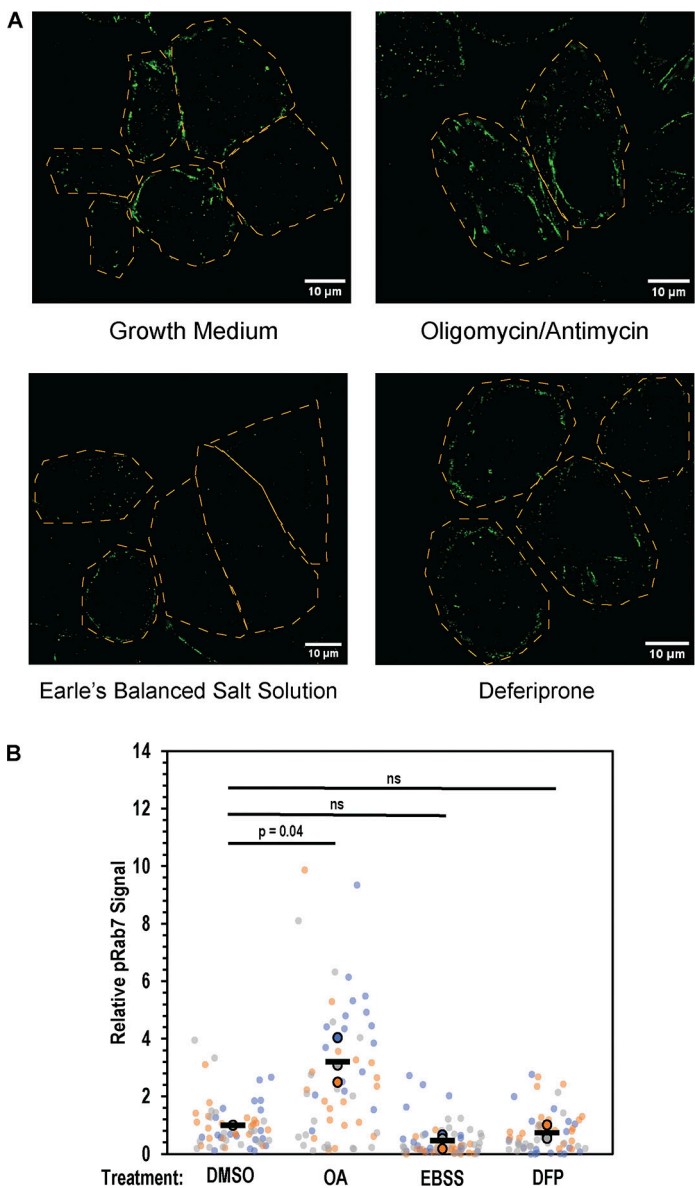

Figure S2. **pS72 Rab7 formation following depolarization, starvation, and iron chelation. (A)** Parkin-expressing HeLa cells were treated for 1 H with either growth medium, growth medium supplemented with oligomycin/antimycin, Earle's Balanced Salt Solution, or growth medium supplemented with deferiprone, fixed, and then labeled using an antibody specific for pS72 Rab7. **(B)** Quantification of pRab7 signal. Images of cells were backgrounded subtracted, cells were masked and total fluorescence intensity was measured on a per cell basis. At least 15 cells were measured in this fashion, and averaged, constituting a single biological replicate (outlined, colored circles). This was repeated for a total of three independent replicates for each condition. Horizontal black bars indicate average of the three replicates. Colors indicate replicates performed on the same day. P value was determined using a paired, one-tailed t test.

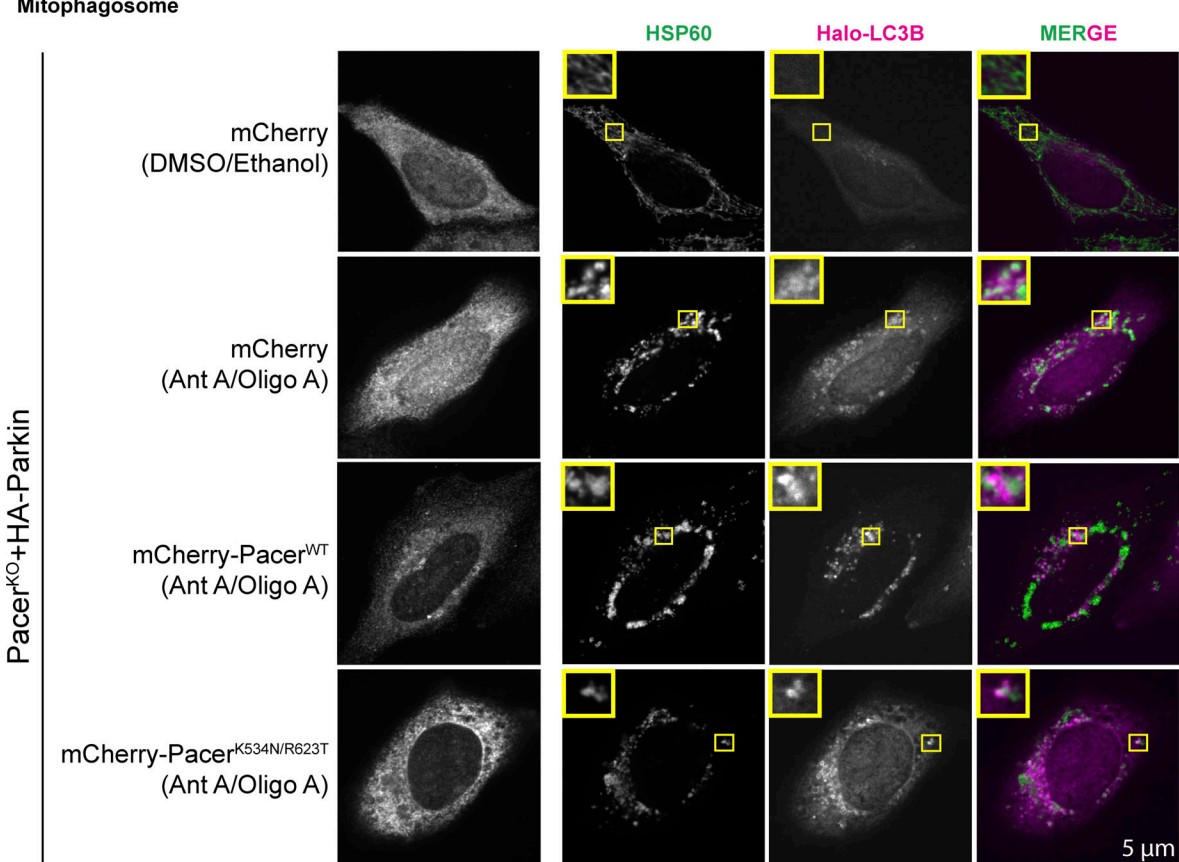

Figure S3. **Immunofluorescence of mitochondria alongside LC3B.** Cells were treated as in Fig. 6 A, fixed, and mitochondria were labeled using the mitochondrial marker HSP60. Under nondepolarizing conditions, LC3B is only diffuse. Treatment with antimycin/oligomycin causes the formation of punctate autophagic structures that are also positive for mitochondria. Insets are 4 µm wide.

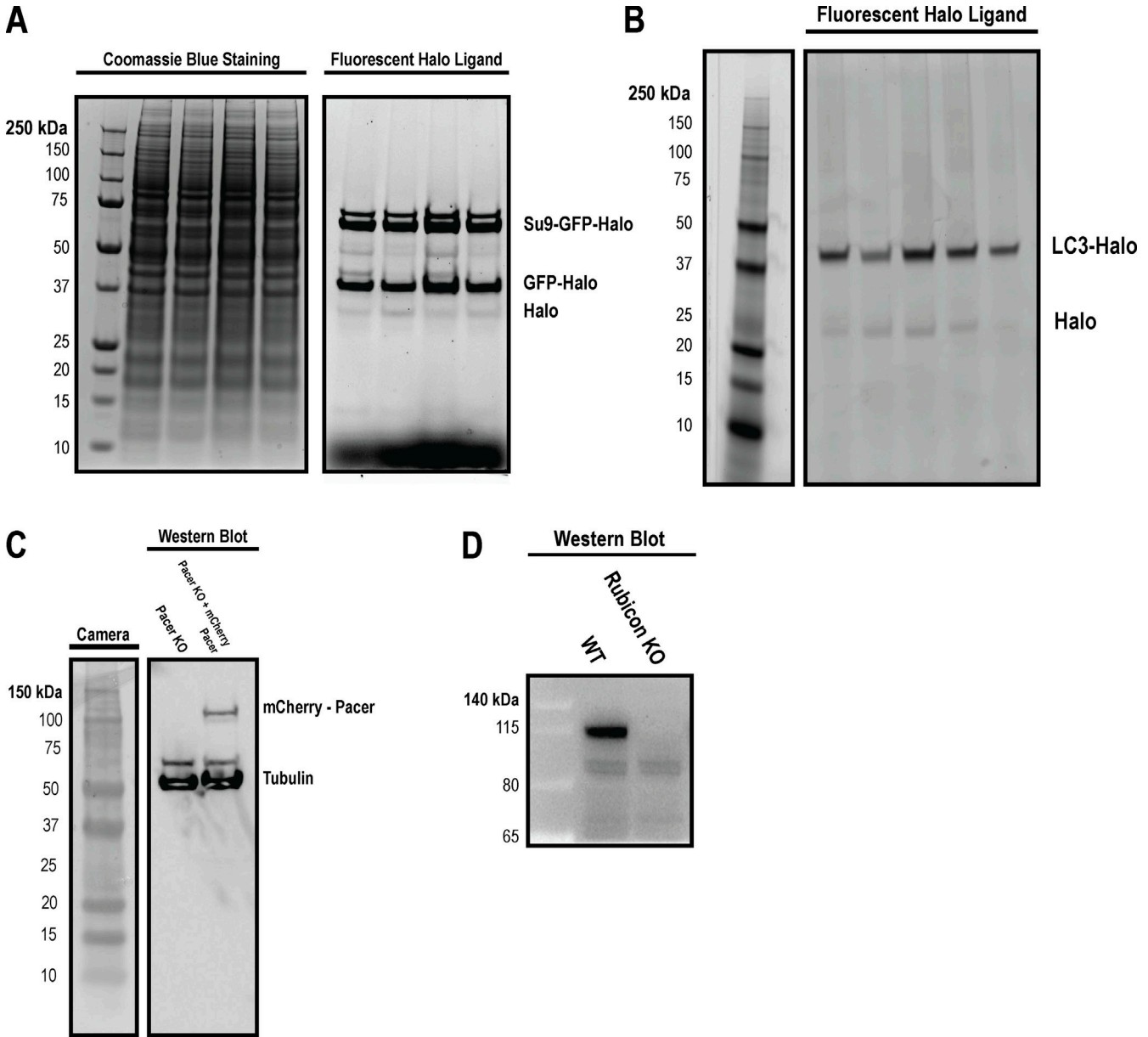

Figure S4. **MW verification of HaloTag flux reporters. (A)** Cells stably expressing Parkin and Su9 HaloTag reporter were labeled with fluorescent halo ligand and then were depolarized with OA for 2 h as indicated in prior experiments. Lysates were run alongside an unstained protein ladder. After acquisition of fluorescent scans, the gel was stained with Coomassie blue dye and imaged to show molecular weight standards. **(B)** Cells stably expressing LC3-Halo were labeled with fluorescent Halo ligand and then were starved for ~1 h as in prior experiments. A prestained molecular weight standard was diluted 1:10, and then 2 µl was run alongside the lysates from this experiment. **(C)** A Western blot was performed against Pacer and Tubulin in Pacer KO and mCherry-Pacer expressing cells as in Fig. 5 C. **(D)** Western blot against WT and Rubicon KO cells with indicated molecular weight markers. Source data are available for this figure: SourceData FS4.

**Provided online are Table S1 and Table S2. Table S1 shows the oligonucleotides for CRISPR KO cell lines. Table S2 shows the details of CRISPR sequences and genotyping results of knockout cell lines in this study.**

