## [Peer Review File · The Journal of Cell Biology]

A RAB7A Phosphoswitch Coordinates Rubicon Homology Protein Regulation of Parkin-Dependent Mitophagy

Dan Tudorica, Bishal Basak, Alexia Puerta Cordova, Grace Khuu, Kevin Rose, Michael Lazarou, Erika Holzbaur, and James Hurley

Corresponding Author(s): James Hurley, University of California, Berkeley

Review Timeline:

Submission Date:	2023-09-05
Editorial Decision:	2023-10-07
Revision Received:	2024-01-12
Editorial Decision:	2024-03-03
Revision Received:	2024-03-13

Monitoring Editor: Jodi Nunnari

Scientific Editor: Dan Simon

Transaction Report:

DOI: <https://doi.org/10.1083/jcb.202309015>

October 7, 2023

Re: JCB manuscript #202309015

Prof. James Hurley
University of California, Berkeley
374d Stanley Hall
Berkeley 94708

Dear Prof. Hurley,

Thank you for submitting your manuscript entitled "A RAB7A Phosphoswitch Coordinates Rubicon Homology Protein Regulation of Parkin-Dependent Mitophagy." Your manuscript has been assessed by expert reviewers, whose comments are appended below. Although the reviewers express potential interest in this work, significant concerns unfortunately preclude publication of the current version of the manuscript in JCB.

You will see that the reviewers feel that the study suggests a new and interesting phosphoswitching mechanism of mitophagy regulation in cells. While the biochemical and structural analyses convincingly support this model, the evidence supporting its validity in cells needs to be substantiated. To address this, the reviewers ask for additional experiments in order to better quantify mitophagy and its characteristics as well as to investigate the molecular mechanisms by which Rubicon and Pacer regulate mitophagy in more detail.

Please let us know if you are able to address the major issues outlined above and wish to submit a revised manuscript to JCB. Note that a substantial amount of additional experimental data likely would be needed to satisfactorily address the concerns of the reviewers. The typical timeframe for revisions is three to four months. While most universities and institutes have reopened labs and allowed researchers to begin working at nearly pre-pandemic levels, we at JCB realize that the lingering effects of the COVID-19 pandemic may still be impacting some aspects of your work, including the acquisition of equipment and reagents. Therefore, if you anticipate any difficulties in meeting this aforementioned revision time limit, please contact us and we can work with you to find an appropriate time frame for resubmission. Please note that papers are generally considered through only one revision cycle, so any revised manuscript will likely be either accepted or rejected.

If you choose to revise and resubmit your manuscript, please also attend to the following editorial points. Please direct any editorial questions to the journal office.

GENERAL GUIDELINES:

Text limits: Character count is < 40,000, not including spaces. Count includes title page, abstract, introduction, results, discussion, and acknowledgments. Count does not include materials and methods, figure legends, references, tables, or supplemental legends.

Figures: Your manuscript may have up to 10 main text figures. To avoid delays in production, figures must be prepared according to the policies outlined in our Instructions to Authors, under Data Presentation, <https://jcb.rupress.org/site/misc/ifora.xhtml>. All figures in accepted manuscripts will be screened prior to publication.

Supplemental information: There are strict limits on the allowable amount of supplemental data. Your manuscript may have up to 5 supplemental figures. Up to 10 supplemental videos or flash animations are allowed. A summary of all supplemental material should appear at the end of the Materials and methods section.

Please note that JCB now requires authors to submit Source Data used to generate figures containing gels and Western blots with all revised manuscripts. This Source Data consists of fully uncropped and unprocessed images for each gel/blot displayed in the main and supplemental figures. Since your paper includes cropped gel and/or blot images, please be sure to provide one Source Data file for each figure that contains gels and/or blots along with your revised manuscript files. File names for Source Data figures should be alphanumeric without any spaces or special characters (i.e., SourceDataF#, where F# refers to the associated main figure number or SourceDataFS# for those associated with Supplementary figures). The lanes of the gels/blots should be labeled as they are in the associated figure, the place where cropping was applied should be marked (with a box), and molecular weight/size standards should be labeled wherever possible. Source Data files will be made available to reviewers during evaluation of revised manuscripts and, if your paper is eventually published in JCB, the files will be directly linked to specific figures in the published article.

If you choose to resubmit, please include a cover letter addressing the reviewers' comments point by point. Please also highlight all changes in the text of the manuscript.

Regardless of how you choose to proceed, we hope that the comments below will prove constructive as your work progresses. We would be happy to discuss them further once you've had a chance to consider the points raised. You can contact the journal office with any questions at cellbio@rockefeller.edu.

Thank you for thinking of JCB as an appropriate place to publish your work.

Sincerely,

Jodi Nunnari, PhD
Editor-in-Chief
Journal of Cell Biology

Dan Simon, PhD
Scientific Editor
Journal of Cell Biology

Reviewer #1 (Comments to the Authors (Required)):

This interesting manuscript documents the Rab binding preferences of Rubicon (RUBCN) compared with the related Pacer (RUBCNL) protein in regulating mitophagy. In previous work, Hurley and colleagues reported the structure of Rab7A with the Rubicon RH domain. Here, in structure models, the authors detect a significant difference in the RH domain Rab7A binding pocket where that of Rubicon binds non-phosphorylated Rab7A and that of PACER binds phosphorylated Rab7A; subcellular localization confirms this distinction and mutagenesis blocks phosphoRab7A association. Pacer knockout HeLa cells showed reduced mitophagy upon depolarization. Finally, the authors conclude that Pacer acts early in mitophagosome expansion because the average area of autophagosomes is higher with wild type PACER compared with mutant.

Readers of JCB will surely be interested in this story. However, the later part of the manuscript could easily be strengthened. For example, in the analysis of PACER's role in increasing autophagosome area, the results should be normalized to PACER expression in each cell. If it acts early, the authors should show kinetics after 1 hour of OA treatment--they should monitor and document a rate of growth. Also, not all the structures in Fig. 6 are mitophagosomes, or? Here the authors need to document the process without OA for comparison and explain it clearly.

Minor:

Figure 5C,D. ECL blots will exaggerate differences between low level proteins due to its non-linearity. Can the authors not use a non-amplifying LICOR for more accurate determinations? It seems that RUBCN and RUNCNL are normally very poorly abundant. Please include some discussion of how that would be adequate to drive the processes studied here and how the switch works in mitophagy for less expert readers.

The authors claim that in HeLa cells Pacer is specific for PINK1/PARKIN mitophagy, in contrast to another report that seems to have used U2OS cells. The authors should either knockdown PACER in U2OS cells or more clearly explain why their data differ from the prior report (ref. 29).

In several figures, the authors analyze multiple lanes on a single gel and appear to use each lane as an N value for statistics. In all cases, please indicate where independent experiments were carried out which would be necessary for proper P value determinations.

Please use a dashed line to outline cells in the cell images in Figs. 3,4,6; line scans are not helpful and less informative than the co-localization quantitation that is included.

Figure 2F legend. Please specify the protein here.

Page 8, Rab phosphorylation has not been widely seen in the endocytic system--only endocytic pathway Rab7 is phosphorylated. Perhaps a better word would be "in the membrane trafficking" system? Please rephrase.

Reviewer #2 (Comments to the Authors (Required)):

The role of RAB7 and its effectors Rubicon and Pacer in autophagy has been well characterized. In this manuscript, Tudorica et al. investigated their functions in mitophagy. They found that TBK1-dependent phosphorylation of RAB7A modulates mitophagy by switching the recruitment of RH domain-containing effectors, Rubicon and Pacer. Non-phosphorylated RAB7A prefers Rubicon binding for mitophagy suppression, while Ser-72 phosphorylated RAB7A (pRAB7A) recruits Pacer to facilitate mitophagy. Structural and mutation analyses showed that a basic triad in the RH domain of Pacer that is absent in Rubicon is responsible for the differential recruitment by pRAB7A and RAB7A. The authors claimed that phosphorylation of RAB7A at Ser-72 acts as a switch to coordinate the opposing function of Rubicon and Pacer in mitophagy. In this study, mitophagy is assessed solely based on immunoblotting assays and the differences are relatively mild. The mechanism by which Pacer regulates mitophagy has not been tackled. Overall, the conclusions drawn in this study are not convincingly substantiated by the experimental data.

Major concerns:

1. Fig. 1C-1D and Fig. 5, the difference in the blotting signals for Halo among different cells is relatively small. Loading controls such as ACTIN or GAPDH are missing in all immunoblotting experiments, which lead to the small difference questionable. Other assays such as fluorescence imaging using specific mitophagy reporters such as Parkin or mt-Keima should be performed to determine mitophagy activity.
2. The interaction between RAB7 and pRAB7A with Rubicon and Pacer should be determined in vivo under different conditions.
3. Fig. 3A, 3D and 4C, high magnification images showing colocalization of Rubicon or Pacer with RAB7A and pRAB7A should be presented. The antibody specificity should be confirmed. The GFP reporter for RAB7A could also be analysed.
4. The mechanism by which Pacer regulates mitophagy should be analysed. RAB7 functions at the step of autophagosome-lysosome fusion. How does phosphorylation of RAB7 modulate the role of Pacer in autophagosome formation?
5. Fig. 6, LC3B puncta represent autophagic structures. Specific reporters for mitophagy such as Parkin or mt-Keima should be examined to determine the formation and turnover of mitophagosomes.
6. The level, distribution and dynamic of pRAB7A and its colocalization with Rubicon and Pacer under normal and mitophagy-induced conditions should be determined.
7. Does Ser-72 phosphorylation of RAB7A act as a switch for the recruitment of Rubicon and Pacer during starvation-induced autophagy?
8. The authors claimed that TBK1-dependent phosphorylation of RAB7A serves as a switch for the recruitment of Rubicon and Pacer. Does TBK1 act at the same step as Pacer/Rubicon in mitophagy?

Minor concerns:

1. Fig. 5A, the blotting signals for Su9-Halo are overexposed.
2. Page 6, lines 15 and 16, the authors have cited wrong figure numbers. "Figure 3A" and "Figure 3B" should be "Figure 3D" and "Figure 3E".
3. Figure S2 is missing.

Reviewer #3 (Comments to the Authors (Required)):

In the manuscript by Tudorica et al., the authors uncover a mechanism whereby the phosphorylation status of Rab7 determines an interaction with Rubicon or Pacer, which they propose to inhibit or activate Parkin-dependent mitophagy respectively. The authors do indeed appear to have uncovered a very elegant phosphoswitching mechanism of mitophagy regulation and I'm sure this work will be of interest to many in the membrane trafficking field in general too. However, while the in vitro work is convincing, the actual mechanism of the pathway in cells is less clear and requires a little more work.

Main points:

- 1) In Figure 1, the mitophagy changes upon loss of Rubicon are very small and not clear by looking directly at the western blot data. Are changes more obvious at different time points, or are there additional mitochondrial markers that can be looked at? Is it possible that the changes are just clonal - for example can the increase in mitophagy be reversed by re-expressing WT Rubicon in the KOs (or exacerbated with the phospho-binding mutant - see point 3 below)?
- 2) In Figure 3 and 4, the authors examine the co-localization of Rubicon and Pacer with Rab7/phospho-Rab7 only following 1h of CCCP. It will be important to compare under basal conditions too. Do phospho-Rab7 levels increase with CCCP and does this result in more co-localization with Pacer and less with Rubicon compared to untreated?
- 3) The authors very nicely identify a set of Rubicon mutants that result in increased phospho-Rab7 binding (Fig. 3D and E - mistakenly referred to as 3A and B in text on page 6). However, the authors do not test the consequences of this in a cellular setting, which would greatly enhance their arguments. For example, if loss of Rubicon binding to phospho-Rab7 is key, then overexpression of the mutant (phospho-binding) Rubicon should have a greater inhibitory effect on mitophagy than overexpression of WT Rubicon (potentially by competing off Pacer). Is this the case?
- 4) In Figure 5 (similar to Figure 1) the changes in mitophagy are not clear - this is especially true with the WT and mutant rescues of Pacer KO, which shows significance only at 2h of depolarization and not at 4 (compared to other treatments at 4h).

Additionally, if we look at relative changes at the significant time point, the amount of reporter being digested is small - going from approx. 4.4% in WT to 3.6% in mutant. While this is significant, is it actually relevant? Some comments on this would be helpful for the readers - is this a minor or major pathway for Parkin-mediated mitophagy?

5) From the data in Fig. 6, the authors argue that Pacer is regulating autophagosome initiation. While I do think the given data supports this hypothesis, more evidence is needed. Does mutant Pacer still interact with the PI3KC3-complex? Is there any observable decrease in VPS34 activity - for example reduced WIPI puncta, which could correlate with initiation changes? Does WT Pacer co-localize with mitochondria or autophagy initiation components (e.g. WIPI) or autophagosomes containing mitochondria? Is this different with mutant Pacer?

Minor point

6) Western blots need MW markers.

We thank all three reviewers for their time and enthusiasm for our study.

Reviewer #1 (Comments to the Authors (Required)):

Readers of JCB will surely be interested in this story. However, the later part of the manuscript could easily be strengthened. For example, in the analysis of PACER's role in increasing autophagosome area, the results should be normalized to PACER expression in each cell.

This is a potentially interesting point, however we were not able to establish any such correlation at the level of resolution attained in the cell imaging. Fortunately, the conclusions of the manuscript do not depend on such interpretations, so we have not pursued this further.

If it acts early, the authors should show kinetics after 1 hour of OA treatment--they should monitor and document a rate of growth.

We examined the formation of WIPI2 puncta after one hour of depolarization as was suggested by reviewer 3 (Figure 7). This assay is more sensitive at early time points and a significant difference was seen. This observation adds to the case that Pacer acts early at an early step in autophagy.

Figure 5 shows the kinetics of starvation autophagy and OA and DFP-induced mitophagy in the Pacer KO and mutant cells. The mitophagy data suggests that Pacer has an observable and significant impact as early as 2 hr after treatment. Digestion of LC3 and mitochondrial proteins occurs downstream of WIPI2 puncta formation, it is expected that these events occur later. These data are drawn from the Halo-tag processing assay used throughout the manuscript.

We replicated the LC3B puncta formation assay shown in figure 6 at 1 hr as shown above. The signal is small at this early point. As expected, we did not observe a meaningful difference in the KO and mutant.

Also, not all the structures in Fig. 6 are mitophagosomes, or? Here the authors need to document the process without OA for comparison and explain it clearly.

To address this, we added an additional supplementary figure, S3 (reproduced here):

Mitophagosome

Immunofluorescence of the mitochondrial marker HSP60 imaged alongside Halo-LC3B shows that, prior to OA treatment, no or few autophagosomes are visible. After treatment, autophagosomes are abundant, and essentially all the autophagosomes are positive for mitochondria. This demonstrates that all the autophagic structures in figure 6 are indeed mitophagosomes.

Minor:

Figure 5C,D. ECL blots will exaggerate differences between low level proteins due to its non-linearity. Can the authors not use a non-amplifying LICOR for more accurate determinations?

ECL was used only to verify stable expression of Pacer, and was not used for the autophagy flux assay. The Halotag processing assay that measures autophagy flux was performed using a fluorescent Halo ligand, not ECL. This allows direct and quantitative measurements at high sensitivity. A full validation of this method was performed in Yim et al (10.7554/eLife.78923). We feel this is the state of the art approach. The clarifying language “*Pacer expression was probed via Western blot, while HaloTag processing was directly imaged using a fluorescent Halo ligand*” has been added to the figure caption of figure 5.

It seems that RUBCN and RUBCNL are normally very poorly abundant. Please include some discussion of how that would be adequate to drive the processes studied here and how the switch works in mitophagy for less expert readers.

Proteomics data indicates that RUBCN is expressed at similar low levels comparable to the PI3KC3 subunits. According to OpenCell, Rubicon is maintained at 8.4 nM in a typical HEK cell, while PI3KC3 subunits UVRAG and ATG14 are present at 22 and 11 nM, respectively. Estimates of copy numbers in autophagy initiation suggest only ~30 molecules of the upstream components are involved in this reaction (Banerjee et al. Sci Adv. 2023). We added the sentence “Here we use “elevated” and “reduced” in a relative sense, as PI3KC3 subunits, Pacer, and Rubicon are all present at low concentrations and copy numbers in cells, in the single digit to low double digit nM levels³¹” at the start of pg. 4.

The authors claim that in HeLa cells Pacer is specific for PINK1/PARKIN mitophagy, in contrast to another report that seems to have used U2OS cells. The authors should either knockdown PACER in U2OS cells or more clearly explain why their data differ from the prior report (ref. 29).

We appreciate that the referee requests either the knockdown experiment or a clearer explanation. We opted for the latter. We suspect experimental rather than cell type differences are involved. Our manuscript benefits from the use of the most sensitive method of measuring autophagy and mitophagy flux, which was not available at the time the work in ref. 29 was carried out. We used an inducible HaloTag system to perform quantitative measurements of autophagy flux. Ref. 29 used confocal-based LC3 GFP/RFP quenching assays as well as Western blots following the degradation of the autophagy cargo receptor p62. These methods indirectly measure autophagy flux via characteristic proteins at particular points within the autophagy pathway. In contrast, the HaloTag processing assay directly measures the proportion of labelled mitochondria that have been delivered to lysosomes at defined time points, enabling more precise determination of mitophagy flux.

In several figures, the authors analyze multiple lanes on a single gel and appear to use each lane as an N value for statistics. In all cases, please indicate where independent experiments were carried out which would be necessary for proper P value determinations.

Clarifying language has been added to the captions of these figures. See highlighted text in figure legends. In all cases, P-values were calculated using independent biological replicates. In certain cases, lysates were retained following multiple days of experimentation, and run on a single gel. As a result, each lane represents an independent experiment.

Please use a dashed line to outline cells in the cell images in Figs. 3,4,6; line scans are not helpful and less informative than the co-localization quantitation that is included.

Dashed lines have been added to all cell images.

Figure 2F legend. Please specify the protein here.

We added “Rubicon” to the figure caption, and added an additional label to the figure to promote clarity.

Page 8, Rab phosphorylation has not been widely seen in the endocytic system-only endocytic pathway Rab7 is phosphorylated. Perhaps a better word would be "in the membrane trafficking" system? Please rephrase.

This point is well-taken, and the language (now highlighted in the text) has been changed to reflect this.

Reviewer #2 (Comments to the Authors (Required)):

The role of RAB7 and its effectors Rubicon and Pacer in autophagy has been well characterized. In this manuscript, Tudorica et al. investigated their functions in mitophagy. They found that TBK1-dependent phosphorylation of RAB7A modulates mitophagy by switching the recruitment of RH domain-containing effectors, Rubicon and Pacer. Non-phosphorylated RAB7A prefers Rubicon binding for mitophagy suppression, while Ser-72 phosphorylated RAB7A (pRAB7A) recruits Pacer to facilitate mitophagy. Structural and mutation analyses showed that a basic triad in the RH domain of Pacer that is absent in Rubicon is responsible for the differential recruitment by pRAB7A and RAB7A. The authors claimed that phosphorylation of RAB7A at Ser-72 acts as a switch to coordinate the opposing function of Rubicon and Pacer in mitophagy. In this study, mitophagy is assessed solely based on immunoblotting assays and the differences are relatively mild. The mechanism by which Pacer regulates mitophagy has not been tackled. Overall, the conclusions drawn in this study are not convincingly substantiated by the experimental data.

Major concerns:

1. Fig. 1C-1D and Fig. 5, the difference in the blotting signals for Halo among different cells is relatively small. Loading controls such as ACTIN or GAPDH are missing in all immunoblotting experiments, which lead to the small difference questionable. Other assays such as fluorescence imaging using specific mitophagy reporters such as Parkin or mt-Keima should be performed to determine mitophagy activity.

The HaloTag based flux assay does not use blotting. Since the HaloTag reporter is fluorescently labelled as the assay is run, fluorescence intensity measurements are made immediately after running the gel, without any of the washing or blotting steps typical to Western blots. A full validation of this method was performed in Yim et al (10.7554/eLife.78923). This is the best available approach. The HaloTag assay is quantified by determining the proportion of HaloTag reporter delivered to the lysosome. The upper band represents undelivered, undigested reporter, while the lower band represents delivered, digested reporter. To quantify flux, we divide the total signal intensity of the lower, digested band, over the total halo signal, to calculate the fraction of the reporter that has been digested by autophagy. As autophagy occurs, the upper band grows

dimmer, and the lower band grows brighter. In other words, each lane has an internal control in the form of the undigested Halo band, rendering the method robust to differences in loading.

2. The interaction between RAB7 and pRAB7A with Rubicon and Pacer should be determined in vivo under different conditions.

Figure 3A shows that wild-type but not mutant Rubicon colocalizes with RAB7. Figure 4C shows that wild-type but not mutant Pacer colocalizes with pRAB7A in cells. A new supplementary figure (Figure S2) shows that pRAB7A puncta formed in response to OA are sparse. This sparseness and the low copy numbers of Rubicon and Pacer deterred us from attempted to demonstrate interaction by coimmunoprecipitation.

3. Fig. 3A, 3D and 4C, high magnification images showing colocalization of Rubicon or Pacer with RAB7A and pRAB7A should be presented. The antibody specificity should be confirmed. The GFP reporter for RAB7A could also be analysed.

Inserts have been added to 3A and 4C showing close-ups. Figure 3D is an in vitro bead binding assay.

The specificity of the pS72 RAB7A antibody has been characterized in the paper for which it was originally produced, Malik et al (<https://doi.org/10.1042/BCJ20200937>) as well as in Metcalfe et al (<https://doi.org/10.1038/s41467-023-40532-2>). Additional antibody validation has also been performed by the supplier. RAB7A was not marked via GFP, but rather by a very commonly used RAB7A antibody (ab137029).

4. The mechanism by which Pacer regulates mitophagy should be analysed. RAB7 functions at the step of autophagosome-lysosome fusion. How does phosphorylation of RAB7 modulate the role of Pacer in autophagosome formation?

We know that pRAB7A forms early in the mitophagy pathway from Hanafusa et al. (10.1242/jcs.260395). This suggests that pRAB7A would have a role in mitophagy initiation, not only in fusion. To further explore this question, we imaged formation of WIPI2 puncta alongside wildtype and mutant Pacer. We find that expression of wildtype Pacer promotes formation of WIPI2 puncta following 1 H of depolarization with oligomycin/antimycin, while mutant Pacer does not (Figure 7A,B). Furthermore, wildtype Pacer robustly colocalizes with these WIPI2 puncta, while mutant Pacer fails to (Figure 7B). This provides direct evidence that Pacer is recruited to sites of mitophagosome formation in a pS72 RAB7A-dependent manner. Taken together, this indicates that RAB7A is phosphorylated at sites of mitophagy initiation, which causes recruitment of Pacer.

5. Fig. 6, LC3B puncta represent autophagic structures. Specific reporters for mitophagy such as Parkin or mt-Keima should be examined to determine the formation and turnover of mitophagosomes.

We show in the new Figure S3 that essentially all LC3B puncta formed in these assays are positive for the mitochondrial marker HSP60, demonstrating that they are mitophagosomes. See response to reviewer 1 on this concern.

6. The level, distribution and dynamic of pRAB7A and its colocalization with Rubicon and Pacer under normal and mitophagy-induced conditions should be determined.

A new supplementary figure has been added that includes a quantitation of pS72 RAB7A formation following a one hour incubation in all the conditions tested in this paper (Figure S3). Here, we show that pS72 RAB7A formation is only increased above baseline in cells treated with the PINK-PARKIN mitophagy inducers oligomycin/antimycin. Similarly, we show that Pacer puncta formation is promoted by treatment with mitochondrial depolarizers (Figure 6B), and that this ability depends on binding to pS72 RAB7A.

7. Does Ser-72 phosphorylation of RAB7A act as a switch for the recruitment of Rubicon and Pacer during starvation-induced autophagy?

In Figure S3 we find that pS72 RAB7A production is only stimulated when PINK1/Parkin mitophagy is activated, consistent with earlier reports that identify pS72 RAB7A as a PINK1/Parkin specific mitophagy initiation factor. Neither starvation nor non-PINK1/Parkin (DFP) mitophagy is able to promote RAB7A phosphorylation. As a result, pS72 RAB7A production is not a switch in starvation autophagy.

8. The authors claimed that TBK1-dependent phosphorylation of RAB7A serves as a switch for the recruitment of Rubicon and Pacer. Does TBK1 act at the same step as Pacer/Rubicon in mitophagy?

Our evidence strongly indicates that this regulatory mechanism acts at the level of initiation, just downstream of TBK1 activation and its phosphorylation of RAB7A. We show using the WIPI2 puncta assay in the new Fig. 7 that Pacer promotes production of PI3P, a pivotal factor in autophagy initiation and that this production depends on Pacer's ability to bind pS72 RAB7A. (Figures 6 and 7).

Minor concerns:

Fig. 5A, the blotting signals for Su9-Halo are overexposed.

These are fluorescence images, not blots. The fluorescent imager that we use automatically detects saturated pixels and marks them, and no saturated pixels were detected in this image.

2. Page 6, lines 15 and 16, the authors have cited wrong figure numbers. "Figure 3A" and "Figure 3B" should be "Figure 3D" and "Figure 3E".

This has been corrected.

3. Figure S2 is missing.

The original Figure S2 referred to here was present in an earlier draft of the paper and was since removed.

Reviewer #3 (Comments to the Authors (Required)):

In the manuscript by Tudorica et al., the authors uncover a mechanism whereby the phosphorylation status of Rab7 determines an interaction with Rubicon or Pacer, which they propose to inhibit or activate Parkin-dependent mitophagy respectively. The authors do indeed appear to have uncovered a very elegant phosphoswitching mechanism of mitophagy regulation and I'm sure this work will be of interest to many in the membrane trafficking field in general too. However, while the in vitro work is convincing, the actual mechanism of the pathway in cells is less clear and requires a little more work.

Main points:

1) In Figure 1, the mitophagy changes upon loss of Rubicon are very small and not clear by looking directly at the western blot data. Are changes more obvious at different time points, or are there additional mitochondrial markers that can be looked at? Is it possible that the changes are just clonal - for example can the increase in mitophagy be reversed by re-expressing WT Rubicon in the KOs (or exacerbated with the phospho-binding mutant - see point 3 below)?

We do not believe this observation is a fluke. It is reproducible and completely consistent with the proposed mechanism. The phosphoinhibition of Rubicon binding to RAB7A explains why the change in mitophagy flux upon Rubicon KO is very small relative to the change in autophagy flux. Our interpretation is that pRAB7A production is a means by which mitophagy can relieve the inhibition imposed by Rubicon, since it prevents Rubicon from binding RAB7A, and RAB7A binding is necessary for full autophagy inhibition. Rubicon KO has a minimal effect on mitophagy because Rubicon is already being inhibited by the conversion of RAB7A to pS72 RAB7A.

2) In Figure 3 and 4, the authors examine the co-localization of Rubicon and Pacer with Rab7/phospho-Rab7 only following 1h of CCCP. It will be important to compare under basal conditions too. Do phospho-Rab7 levels increase with CCCP and does this result in more co-localization with Pacer and less with Rubicon compared to untreated?

Indeed, depolarization of mitochondria is a specific promoter of pRAB7A formation. Under nondepolarizing conditions, pS72 RAB7A puncta are quite sparse (figure S2), consistent with earlier reports (Fujita et al, 10.1242/jcs.260395). We also show that the number of Pacer puncta

per cell increases by 50% when treated with mitochondrial depolarizers (Figure 6B), indicating that Pacer is recruited to these novel pS72 RAB7A puncta.

3) The authors very nicely identify a set of Rubicon mutants that result in increased phospho-Rab7 binding (Fig. 3D and E - mistakenly referred to as 3A and B in text on page 6). However, the authors do not test the consequences of this in a cellular setting, which would greatly enhance their arguments. For example, if loss of Rubicon binding to phospho-Rab7 is key, then overexpression of the mutant (phospho-binding) Rubicon should have a greater inhibitory effect on mitophagy than overexpression of WT Rubicon (potentially by competing off Pacer). Is this the case?

This is an attractive idea indeed. However, Rubicon requires binding to unphosphorylated RAB7A in order to maximally inhibit autophagy. While the mutant described here does indeed bind pS72 RAB7A, it is impaired in its ability to bind unphosphorylated RAB7A. As a result, this would not be a clean gain of function mutation, rendering interpretation of experiments using this mutant challenging.

4) In Figure 5 (similar to Figure 1) the changes in mitophagy are not clear - this is especially true with the WT and mutant rescues of Pacer KO, which shows significance only at 2h of depolarization and not at 4 (compared to other treatments at 4h). Additionally, if we look at relative changes at the significant time point, the amount of reporter being digested is small - going from approx. 4.4% in WT to 3.6% in mutant. While this is significant, is it actually relevant? Some comments on this would be helpful for the readers - is this a minor or major pathway for Parkin-mediated mitophagy?

At 4 H, rescue with wildtype Pacer is characterized by a significance score of $p = 0.022$. Given that each data point represents the assay performed on a separate day, it is to be expected that p-values may end up in this neighborhood due to the lower throughput of this approach. Nevertheless, this statistic incorporates the full range of experimental variation in this assay.

The percent turnover of the mitophagy reporter that we found is typical of this assay. Examples in the original methods paper that established this technique reproducibly tracked percent turnover as low as 2% (<https://doi.org/10.7554/eLife.78923>). The assay is able to provide this information because it follows the formation of a lower MW band starting from zero intensity. This makes it more sensitive than traditional mitophagy and autophagy flux assays, which may track the disappearance of a protein relative to a large starting value.

It is also important that the low level of reporter turnover represents a strength of the assay rather than a weakness. Since destruction and recycling of mitochondria can be cytotoxic, it is important to treat the cells as gently as possible. Inducing mitophagy of a large fraction of a cell's mitochondrial would not be physiological, and would not approximate the role of

mitophagy as a quality control system. We believe that our approach allows us to provide data that is as close to native mitophagy as possible.

5) From the data in Fig. 6, the authors argue that Pacer is regulating autophagosome initiation. While I do think the given data supports this hypothesis, more evidence is needed. Does mutant Pacer still interact with the PI3KC3-complex? Is there any observable decrease in VPS34 activity - for example reduced WIPI puncta, which could correlate with initiation changes? Does WT Pacer co-localize with mitochondria or autophagy initiation components (e.g. WIPI) or autophagosomes containing mitochondria? Is this different with mutant Pacer?

To address these questions, we imaged formation of WIPI2 puncta, which are formed downstream of PI3KC3 activation and PI3P production, in cells expressing wildtype and mutant Pacer. We find that expression of wildtype Pacer promotes formation of WIPI2 puncta following 1 H of depolarization with oligomycin/antimycin, while mutant Pacer does not (Figure 7A). Furthermore, wildtype Pacer robustly colocalizes with these WIPI2 puncta, while mutant Pacer fails to (Figure 7B). This provides direct evidence that Pacer is recruited to sites of mitophagosome formation in a pS72 RAB7A- and PI3P-dependent manner.

Minor point

6) Western blots need MW markers.

The HaloTag blots are incompatible with molecular weight markers, since the dyes used in these markers are fluorescent and this tends to drown out the HaloTag signal.

March 3, 2024

RE: JCB Manuscript #202309015R

Prof. James Hurley
University of California, Berkeley
374d Stanley Hall
Berkeley 94708

Dear Prof. Hurley,

Thank you for submitting your revised manuscript entitled "A RAB7A Phosphoswitch Coordinates Rubicon Homology Protein Regulation of Parkin-Dependent Mitophagy." We would be happy to publish your paper in JCB pending final revisions necessary to address the remaining reviewer comments and to meet our formatting guidelines (see details below).

As we discussed please also add a new supplementary figure showing a sample gel with all of the Halo-tagged proteins used in the paper and then stained with Coomassie to visualize molecular weight markers.

A. MANUSCRIPT ORGANIZATION AND FORMATTING:

1) Text limits: Character count for Articles is < 40,000, not including spaces. Count includes title page, abstract, introduction, results, discussion, and acknowledgments. Count does not include materials and methods, figure legends, references, tables, or supplemental legends.

2) Figure formatting: Articles may have up to 10 main text figures. Scale bars must be present on all microscopy images, including inset magnifications. Molecular weight or nucleic acid size markers must be included on all gel electrophoresis. Please add scale bars to Figures 2D, 3D, 4A, and the inset magnifications in 3A, 4C, 5A, & S3. Please label the MW markers on blots in Figure S1A.

Also, please avoid pairing red and green for images and graphs to ensure legibility for color-blind readers. If red and green are paired for images, please ensure that the particular red and green hues used in micrographs are distinctive with any of the colorblind types. If not, please modify colors accordingly or provide separate images of the individual channels.

3) Statistical analysis: Error bars on graphic representations of numerical data must be clearly described in the figure legend. The number of independent data points (n) represented in a graph must be indicated in the legend. Please, indicate whether 'n' refers to technical or biological replicates (i.e. number of analyzed cells, samples or animals, number of independent experiments). If independent experiments with multiple biological replicates have been performed, we recommend using distribution-reproducibility SuperPlots (please see Lord et al., JCB 2020) to better display the distribution of the entire dataset, and report statistics (such as means, error bars, and P values) that address the reproducibility of the findings.

Statistical methods should be explained in full in the materials and methods. For figures presenting pooled data the statistical measure should be defined in the figure legends. Please also be sure to indicate the statistical tests used in each of your experiments (both in the figure legend itself and in a separate methods section) as well as the parameters of the test (for example, if you ran a t-test, please indicate if it was one- or two-sided, etc.). Also, if you used parametric tests, please indicate if the data distribution was tested for normality (and if so, how). If not, you must state something to the effect that "Data distribution was assumed to be normal but this was not formally tested."

4) Materials and methods: Should be comprehensive and not simply reference a previous publication for details on how an experiment was performed. Please provide full descriptions (at least in brief) in the text for readers who may not have access to referenced manuscripts. The text should not refer to methods "...as previously described." Please also describe the acquisition and quantification methods for gels and western blots.

5) For all cell lines, vectors, constructs/cDNAs, etc. - all genetic material: please include database / vendor ID (e.g., Addgene, ATCC, etc.) or if unavailable, please briefly describe their basic genetic features, even if described in other published work or gifted to you by other investigators (and provide references where appropriate). Please be sure to provide the sequences for all of your oligos: primers, si/shRNA, RNAi, gRNAs, etc. in the materials and methods. You must also indicate in the methods the

source, species, and catalog numbers/vendor identifiers (where appropriate) for all of your antibodies, including secondary. If antibodies are not commercial, please add a reference citation if possible.

6) Microscope image acquisition: The following information must be provided about the acquisition and processing of images:

- a. Make and model of microscope
- b. Type, magnification, and numerical aperture of the objective lenses
- c. Temperature
- d. Imaging medium
- e. Fluorochromes
- f. Camera make and model
- g. Acquisition software
- h. Any software used for image processing subsequent to data acquisition. Please include details and types of operations involved (e.g., type of deconvolution, 3D reconstitutions, surface or volume rendering, gamma adjustments, etc.).

7) References: There is no limit to the number of references cited in a manuscript. References should be cited parenthetically in the text by author and year of publication. Abbreviate the names of journals according to PubMed.

8) Supplemental materials: Articles may have up to 5 supplemental figures and 10 videos. Please also note that tables, like figures, should be provided as individual, editable files. A summary of all supplemental material should appear at the end of the Materials and methods section. Please include one brief sentence per item.

9) eTOC summary: A ~40-50 word summary that describes the context and significance of the findings for a general readership should be included on the title page. The statement should be written in the present tense and refer to the work in the third person. It should begin with "First author name(s) et al..." to match our preferred style.

10) Conflict of interest statement: JCB requires inclusion of a statement in the acknowledgements regarding competing financial interests. If no competing financial interests exist, please include the following statement: "The authors declare no competing financial interests." If competing interests are declared, please follow your statement of these competing interests with the following statement: "The authors declare no further competing financial interests."

11) A separate author contribution section is required following the Acknowledgments in all research manuscripts. All authors should be mentioned and designated by their first and middle initials and full surnames. We encourage use of the CRediT nomenclature (<https://casrai.org/credit/>).

12) ORCID IDs: ORCID IDs are unique identifiers allowing researchers to create a record of their various scholarly contributions in a single place. Please note that ORCID IDs are required for all authors. At resubmission of your final files, please be sure to provide your ORCID ID and those of all co-authors.

13) JCB requires authors to submit Source Data used to generate figures containing gels and Western blots with all revised manuscripts. This Source Data consists of fully uncropped and unprocessed images for each gel/blot displayed in the main and supplemental figures. Since your paper includes cropped gel and/or blot images, please be sure to provide one Source Data file for each figure that contains gels and/or blots along with your revised manuscript files. File names for Source Data figures should be alphanumeric without any spaces or special characters (i.e., SourceDataF#, where F# refers to the associated main figure number or SourceDataFS# for those associated with Supplementary figures). The lanes of the gels/blots should be labeled as they are in the associated figure, the place where cropping was applied should be marked (with a box), and molecular weight/size standards should be labeled wherever possible. Source Data files will be directly linked to specific figures in the published article.

14) Journal of Cell Biology now requires a data availability statement for all research article submissions. These statements will be published in the article directly above the Acknowledgments. The statement should address all data underlying the research presented in the manuscript. Please visit the JCB instructions for authors for guidelines and examples of statements at (<https://rupress.org/jcb/pages/editorial-policies#data-availability-statement>).

B. FINAL FILES:

Thank you for your attention to these final processing requirements. Please contact the journal office with any questions at cellbio@rockefeller.edu.

Thank you for this interesting contribution, we look forward to publishing your paper in Journal of Cell Biology.

Sincerely,

Jodi Nunnari, PhD
Editor-in-Chief
Journal of Cell Biology

Dan Simon, PhD
Scientific Editor
Journal of Cell Biology

Reviewer #1 (Comments to the Authors (Required)):

The authors have adequately addressed my concerns. I just request that they add the responses to some of the reviewer points in the revised manuscript (for example, explain where something is a blot or not or internally controlled) because a reader may have the same concerns as a reviewer.

Reviewer #2 (Comments to the Authors (Required)):

My concerns have been addressed

Reviewer #3 (Comments to the Authors (Required)):

The authors have addressed my concerns but I do disagree with a very minor point (which does not need to be addressed further here) that:

"The HaloTag blots are incompatible with molecular weight markers, since the dyes used in these markers are fluorescent and this tends to drown out the HaloTag signal."

Surely the authors can titrate down the fluorescent markers such that they do not overwhelm the sample signal. Alternatively, just use regular prestained markers as most imaging systems will allow you take a normal photo, which can then be overlaid over the fluorescent image to give MWs.

We have responded to all of the editorial and formatting points. The new Fig. S4 provides the requested MW markers relative to the Halo construct.